# Genome-wide RNA polymerase stalling shapes the transcriptome during aging

**Akos Gyenis**[1,2,6], **Jiang Chang**[1,6], **Joris J. P. G. Demmers**[1], **Serena T. Bruens**[1], **Sander Barnhoorn**[1], **Renata M. C. Brandt** [1], **Marjolein P. Baar**[1,3], **Marko Raseta**[1], **Kasper W. J. Derks**[1,4], **Jan H. J. Hoeijmakers** [1,2,5] & **Joris Pothof** [1] ✉

Gene expression profiling has identified numerous processes altered in aging, but how these changes arise is largely unknown. Here we combined nascent RNA sequencing and RNA polymerase II chromatin immunoprecipitation followed by sequencing to elucidate the underlying mechanisms triggering gene expression changes in wild-type aged mice. We found that in 2-year-old liver, 40% of elongating RNA polymerases are stalled, lowering productive transcription and skewing transcriptional output in a gene-length-dependent fashion. We demonstrate that this transcriptional stress is caused by endogenous DNA damage and explains the majority of gene expression changes in aging in most mainly postmitotic organs, specifically affecting aging hallmark pathways such as nutrient sensing, autophagy, proteostasis, energy metabolism, immune function and cellular stress resilience. Age-related transcriptional stress is evolutionary conserved from nematodes to humans. Thus, accumulation of stochastic endogenous DNA damage during aging deteriorates basal transcription, which establishes the age-related transcriptome and causes dysfunction of key aging hallmark pathways, disclosing how DNA damage functionally underlies major aspects of normal aging.

Aging is characterized by progressive molecular, cellular and physiological decline resulting in reduced vitality, age-related diseases and increased mortality. Because many processes decline or are altered with age[1], surprisingly little is known about the functional status of the basal transcription process in aging. Aged rat and fruit fly brains produce fewer messenger RNAs[2,3] and cell-to-cell variation in transcription is increased in several tissues[4–6], while gene-to-gene transcriptional coordination is decreased in aging[7]. However, transcription in aging is mainly studied in relation to gene expression changes. Transcriptomics significantly contributed to the identification of numerous cellular pathways and processes affected in aging[8–10]. Although part of age-related, organ-specific gene expression changes can be explained by transcription factors, microRNAs[11,12], altered cell type composition[8,13] and epigenetic changes[14,15], a recent transcriptomics meta-analysis indicated that most gene expression similarities between aged mouse organs could not be attributed to these known regulatory mechanisms[8].

DNA damage accumulation has been postulated as an underlying cause of normal aging[16,17] and the aforementioned transcriptional phenotypes[6,7,18,19], mainly based on similarities to cells exposed to DNA-damaging agents or premature aging DNA repair disorders such as Cockayne syndrome and trichothiodystrophy. These conditions have defects in transcription-coupled repair (TCR), which leads to stalled RNA polymerases on DNA lesions[20], suggesting that transcription-blocking DNA damage could also be involved in normal

[1]Department of Molecular Genetics, Erasmus MC Cancer Institute, Erasmus University Medical Center, Rotterdam, The Netherlands. [2]University of Cologne, Faculty of Medicine, Cluster of Excellence for Aging Research, Institute for Genome Stability in Ageing and Disease, Cologne, Germany. [3]Center for Molecular Medicine, University Medical Center Utrecht, Utrecht, The Netherlands. [4]Department of Clinical Genetics and School for Oncology & Developmental Biology, Maastricht University Medical Center, Maastricht, The Netherlands. [5]Princess Maxima Center for Pediatric Oncology, Oncode Institute, Utrecht, The Netherlands. [6]These authors contributed equally: Akos Gyenis, Jiang Chang. ✉e-mail: j.pothof@erasmusmc.nl

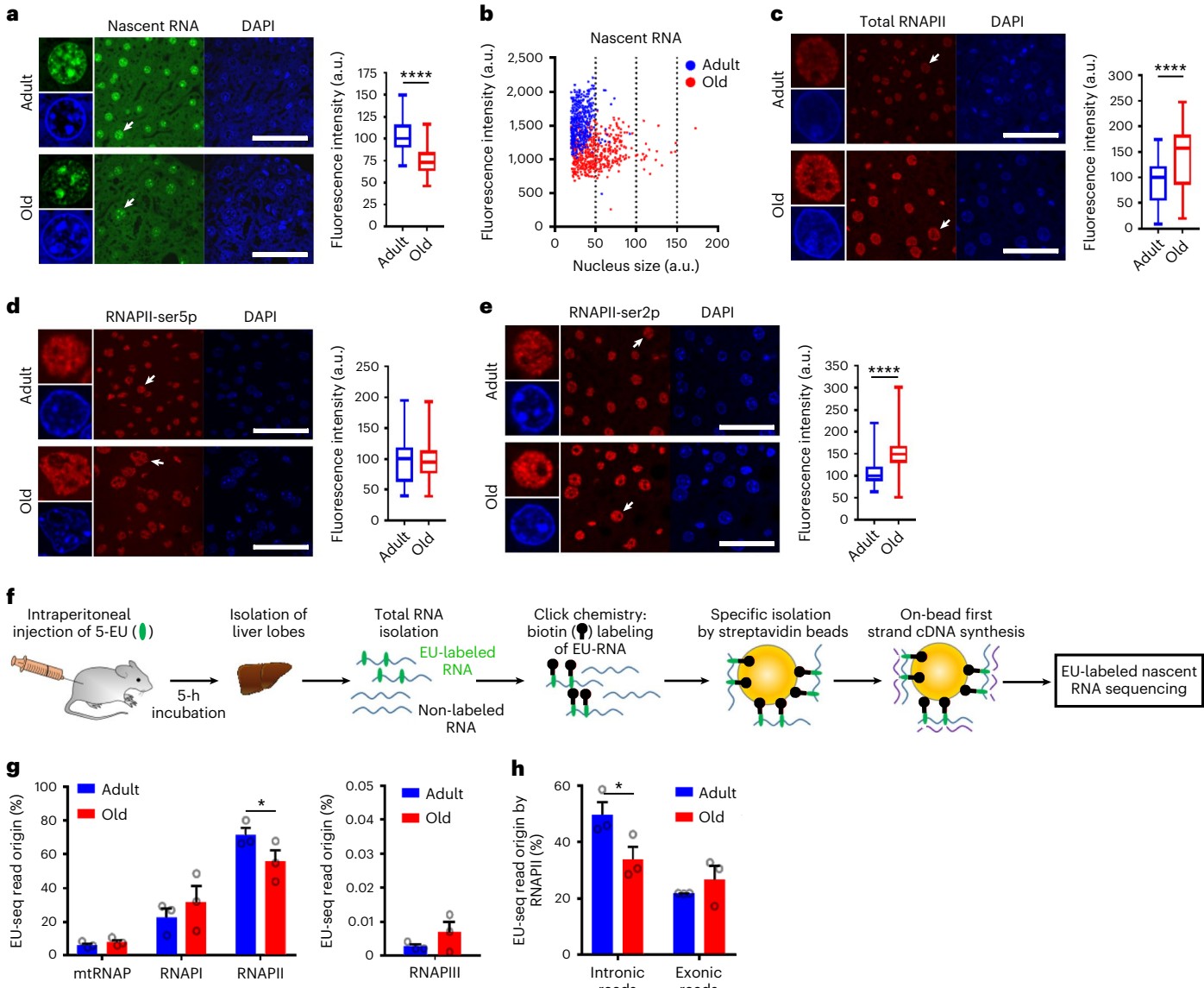

**Fig. 1 | Reduced RNA synthesis and increased RNAPII levels in aged liver.**
**a**, EU-labeled nascent RNA (green) in hepatocyte nuclei (DAPI counterstain, blue) in adult (blue) and old mouse liver (red). Right, Fluorescence intensities quantified in box and whisker plots. The center lines show the medians, the box limits mark the IQR, and the whiskers indicate the minimum and maximum values. $P = 2.1129 \times 10^{-129}$ (two-sided unpaired *t*-test). Counted nuclei: adult $n = 506$; old $n = 500$; $n = 3$ mice per group. **b**, XY scatterplot of fluorescence intensity of EU-labeled nascent RNA (arbitrary units (a.u.)) and corresponding nuclear sizes measured in individual hepatocytes of WT adult (blue) and old (red) liver. **c**–**e**, Total RNAPII (**c**), RNAPII phosphorylated at ser5p (**d**) and RNAPII phosphorylated at ser2p (**e**) immunofluorescence staining (red) in hepatocytes (counterstained by DAPI, blue) in adult and old liver. Box and whisker plots of fluorescence intensities. The center lines show the medians, the box limits mark the IQR, and the whiskers indicate the minimum and maximum values. *P* values by two-sided unpaired *t*-test, $n = 3$ mice per group. Counted nuclei and *P* values: **c**, adult: $n = 206$; old: $n = 155$, $P = 6.64186 \times 10^{-21}$; **d**, adult $n = 2,926$; old $n = 2,643$, $P = 0.323195587$; **e**, adult $n = 2,697$; old $n = 2,708$. $P = 0$. Scale bar, 50 μm. **f**, Flow chart of the experimental procedure for EU-labeled nascent RNA sequencing. **g**, Fraction (%) of EU-seq reads synthesized by different RNA polymerases. RNAPI–II and mtRNAP (left) and RNAPIII (right), with total sequence reads of adult and old normalized to 100%. Data are the mean ± s.e.m. $n = 3$ mice per group. $P = 0.012868073$ (two-sided unpaired *t*-test). **h**, Fraction (%) of EU-seq reads by RNAPII from intronic and exonic regions. Data are the mean ± s.e.m. $n = 3$ mice per group. $P = 0.013520897$ (two-sided unpaired *t*-test).

aging. Although endogenous transcription-blocking DNA lesions accumulate in normal aging[21–25], it is currently not clear whether they elicit significant transcriptional responses. In this study, we analyzed the basal transcription underlying gene expression changes in normal wild-type (WT) aged mice using an in vivo nascent RNA sequencing method combined with RNA polymerase II (RNAPII) chromatin immunoprecipitation followed by sequencing (ChIP–seq) and confocal imaging. We reveal a strong age-related transcriptional decline and skewing of transcriptional output by accumulating DNA damage as a general aging phenotype, causing age-related transcription changes in general, particularly affecting life span-determining aging hallmark pathways.

## Results

### RNAPII transcription is altered in aging liver

To investigate the process of transcription in normal aging, adult (15 weeks) and aged (2 years) WT male mice ($n = 3$ per group) received a single intraperitoneal injection with ethynyl-uridine (EU), a uridine analog that is incorporated into newly synthesized RNA in vivo[26]. Five hours after injection, fluorescence staining of EU revealed a

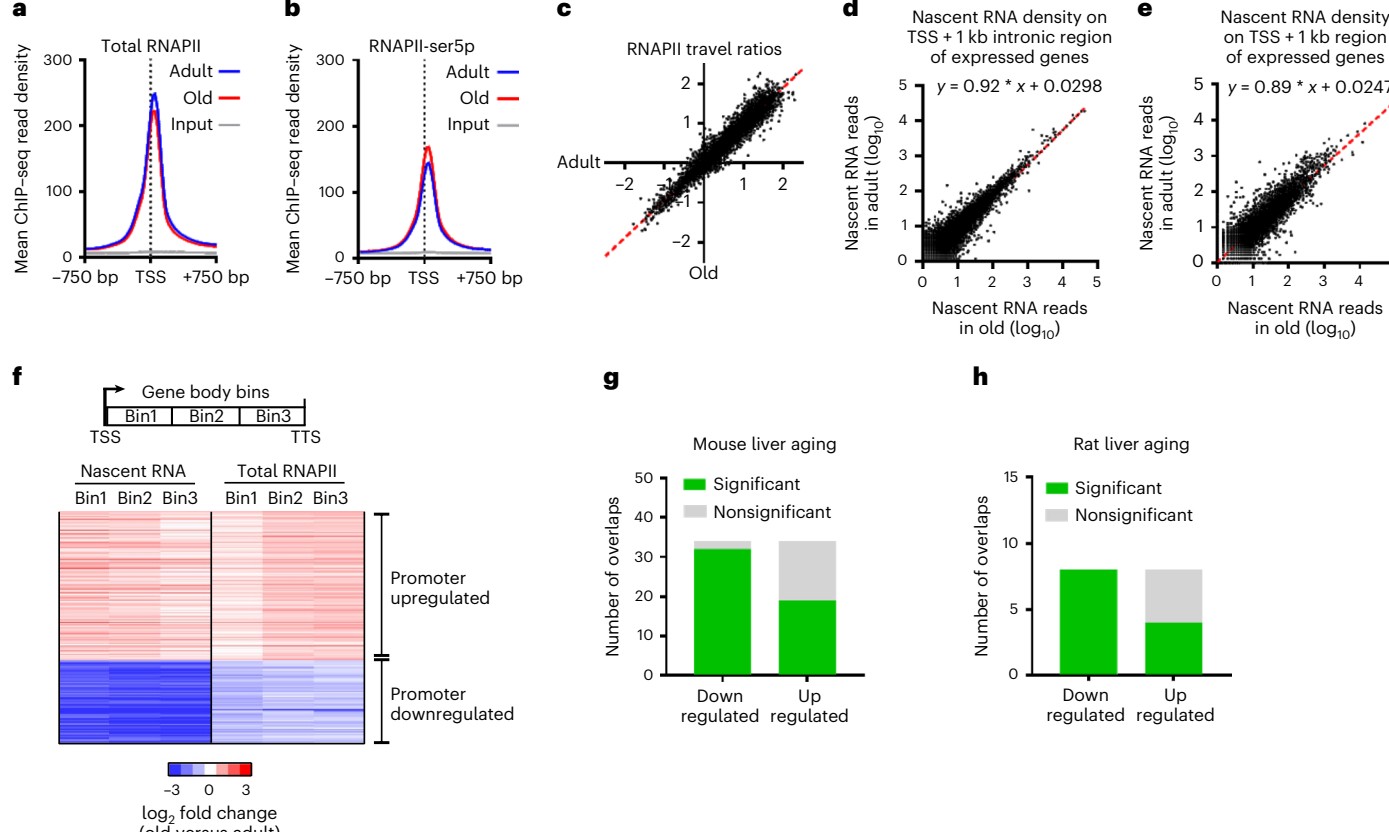

**Fig. 2 | RNAPII promoter activity in aged liver. a,b,** Mean total RNAPII and RNAPII ser5p ChIP–seq read abundance around TSS (TSS ± 750 bp region) of all genes in adult (blue) and old (red) livers. The gray line represents input DNA control ChIP–seq. **c,** XY scatterplot of RNAPII travel ratio of all expressed genes from adult (*x* axis) and old (*y* axis) liver in total RNAPII ChIP–seq data. Each dot represents a gene. Each gene is the average of *n* = 3 mice per group. **d,e,** XY scatterplot depicting nascent RNA synthesis the first 1 kb of introns from the TSS (**d**) or from the TSS to 1 kb downstream (**e**) of all genes in adult (*y* axis) and old (*x* axis) livers. Each dot represents a gene in which the signal represents the mean of *n* = 3 mice. **f,** Three-bin heatmap of log₂ fold changes (old/adult) of nascent RNA (left) and total RNAPII (right) on gene bodies of promoter-upregulated and downregulated clusters genes. Each row represents one gene. **g,h,** Bar diagram showing the overlap between all GSEA aging datasets from mice (**g**) or rat (**h**) and the transcriptionally upregulated and downregulated clusters. The significance and FDR for each overlap were calculated by Fisher's exact test and multiple testing correction by Benjamini–Hochberg method. FDR < 0.05 defined as significant.

1.5-fold-reduced EU signal in old livers (Fig. 1a). The decrease was liver-wide, affecting nearly all hepatocytes and was not restricted to age-related polyploidization (Fig. 1b). Because the reduction of EU signal was pan-nuclear, except for nucleoli (Fig. 1a), pointing to reduced RNAPII-dependent transcription, we tested whether lower RNAPII levels could explain the reduced transcription. Surprisingly, immunofluorescence staining of RNAPII using the same liver samples indicated a 1.4-fold increase, rather than decrease in aged liver (Fig. 1c and Extended Data Fig. 1a). RNAPII initiation and promoter proximal pausing as marked by phosphorylation of serine 5 residues (ser5p) in the C-terminal domain (CTD) did not significantly differ (Fig. 1d and Extended Data Fig. 1b), suggesting that genome-wide RNAPII promoter activity is largely unaltered in aging. However, elongating RNAPII marked by serine 2 CTD phosphorylation (ser2p) demonstrated a 1.5-fold increase (Fig. 1e and Extended Data Fig. 1c). These data indicate that basal transcription is altered in aged liver.

To examine these seemingly conflicting observations of reduced transcription and increased RNAPII abundance, we selectively isolated and sequenced in vivo EU-labeled nascent RNA (EU-seq), which resulted in a significant higher proportion of intronic reads compared to total RNA sequencing (Fig. 1f and Extended Data Fig. 2a–c). Control experiments pointed to identical EU incorporation densities in adult and old livers (Extended Data Fig. 2d–g), ruling out lower EU uptake as the explanation for the lower EU signal in old liver. Next,

we determined the contribution of each RNA polymerase to the cellular nascent RNA pool by assigning reads to RNA species transcribed by each of the different polymerases. As expected, the majority of EU-labeled RNA originated from RNAPII (Fig. 1g and Extended Data Fig. 2h), the only RNA polymerase displaying a significant age-related reduction in RNA synthesis as also apparent from the approximately 1.5-fold decrease in intron-derived sequence reads (Fig. 1h). As splicing events were not significantly altered (Extended Data Fig. 2i,j), the disparity between reduced de novo RNA synthesis and increased elongating RNAPII suggests a specific lower RNAPII productivity in aging.

### Genome-wide promoter activity is normal in aging

To further examine the discrepancy between RNAPII abundance and transcription, we performed ChIP–seq using antibodies against total, ser5p and ser2p RNAPII from the same livers described above. We first investigated whether genome-wide promoter silencing could explain the reduced transcription. In agreement with the immunofluorescence results (Fig. 1), total and ser5p RNAPII occupancy genome-wide at transcriptional start sites (TSS) across all genes did not significantly differ in aging (Fig. 2a,b). Also, the transition of RNAPII from promoter to productive elongation was unaltered (Fig. 2c). To assess transcription proceeding into early elongation, we measured genome-wide nascent RNA production in the first kilobase as measured in the first kilobase

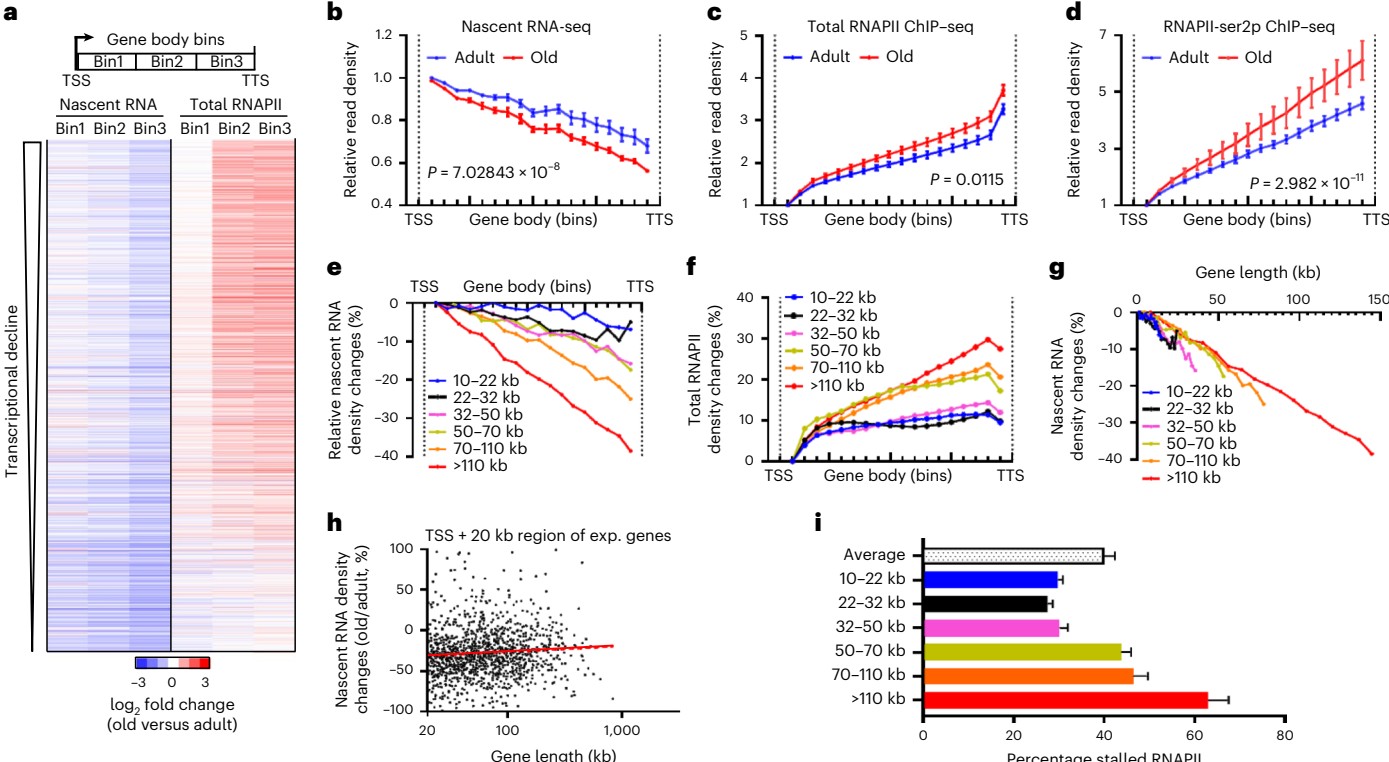

**Fig. 3 | Gene-length-dependent RNAPII stalling in old mouse livers. a**, Three-bin heatmap of log$_2$ fold changes (old/adult) of nascent RNA and ChIP–seq of total RNAPII on gene bodies, sorted by level of transcriptional decline on all expressed genes. **b–d**, Relative sequencing densities of the transcription elongation phase between TSS and TTS. **b**, Nascent RNA sequencing in adult (blue) and aged (red) liver. **c**, Total RNAPII ChIP–seq in adult (blue) and aged (red) liver. **d**, RNAPII-ser2p ChIP–seq in adult (blue) and aged (red) liver. All expressed genes; *n* = 3 mice per group. *P* values: unpaired two-sided *t*-test with 32 d.f. Data are presented as the mean ± s.e.m. **e,f**, Percentage sequencing read density change in the transcription elongation phase in aging between TSS and TTS of gene categories based on genomic gene length: EU-seq (**e**) and total RNAPII ChIP–seq (**f**). **g**, Percentage sequencing read density change (old/adult) in EU-seq as seen in Fig. 2e, in which the *x* axis is the average gene length of each category. **h**, Scatter plot depicting all expressed gene lengths >20 kb (*x* axis) and percentage change between old and adult in EU-seq densities from TSS to 20 kb downstream (*y* axis) (*n* = 3,308). **i**, Percentage stalled RNAPII in gene bodies. The colors indicate the gene-length classes as in Fig. 2e. Data are the mean ± s.d. (10–22 kb: *n* = 662; 22–30 kb: *n* = 644; 30–50 kb: *n* = 788; 50–70 kb: *n* = 587; 70–110 kb: *n* = 643; and >110 kb: *n* = 646).

of intronic regions (Fig. 2d) or from the TSS (Fig. 2e). We observed an almost 1:1 correlation in the first kilobase of transcription across all genes, indicating that overall promoter activity effectively proceeding into transcription is largely unchanged. While genome-wide reduced promoter activity could not explain the reduced transcription phenotype, we expected altered transcription by higher or lower promoter activity. All expressed genes (*n* = 3,970) that represent >90% of RNAPII-dependent nascent RNA production were split into 3 equal bins from its TSS to the transcription termination site (TTS) and corresponding reads from nascent RNA and of RNAPII ChIP–seq mapped in each bin were compared between old and adult liver. Using clustering analysis, we identified genes that were transcriptionally upregulated or downregulated in aging over all bins both in nascent RNA and RNAPII ChIP–seq (Fig. 2f and Extended Data Fig. 3a–c), which reflect promoter regulation. To analyze whether the identified transcriptionally upregulated (*n* = 778) or downregulated (*n* = 394) genes are biologically relevant for aging, we used the Enrichr tool for gene set enrichment analysis (GSEA)[27,28] to compare these gene signatures with the published aging perturbation database containing 34 mouse liver and 15 rat liver mRNA expression profiles. The transcriptionally upregulated and downregulated gene signatures closely resembled the published rodent liver aging profiles (Fig. 2g,h), indicating that promoter regulatory programs during aging are conserved across transcriptomics studies. In summary, the approximately 1.5-fold lower nascent RNA synthesis in old liver is not due to reduced promoter activity or RNAPII transition to elongation.

## Gene-length-dependent stalling of transcription elongation

Close inspection of the three-bin heatmap revealed a consistent pattern of gradually declining nascent transcription across bins over gene bodies in the transcriptionally upregulated genes, whereas RNAPII levels displayed the opposite trend (Fig. 2f). To assess the generality of this phenomenon, we extended the three-bin heatmap to all expressed genes sorted according to the degree of transcriptional decline. Interestingly, almost all genes experienced gradually declining transcription in aging liver and concomitantly increasing RNAPII occupancy throughout gene bodies, disclosing this as a genome-wide phenomenon (Fig. 3a). Promoter-upregulated genes in aging also exhibited this transcriptional decline independent of promoter regulation (Fig. 2f and Extended Data Fig. 3b,c). To better quantify nascent RNA and elongating RNAPII behavior, all expressed genes were divided into 20 bins from TSS to TTS. To exclude reads mapping to the TSS and TTS, we only analyzed bins 2–19, which represent elongation. As expected, we observed an age-independent general gradual decline in nascent RNA across all expressed gene bodies (Fig. 3b), because of the directional nature of transcription, and sequencing complete (growing) nascent RNA molecules and not only the RNAPII footprint. While transcription in the first kilobase of gene bodies is similar (Fig. 2d,e), the decline over the entire genes was significantly stronger in old liver (Fig. 3b). We termed this age-related excess drop in transcription 'gradual loss of productive transcription' (GLPT). In contrast, total and ser2p RNAPII levels gradually increased in gene bodies during aging (Fig. 3c,d), which is consistent with Fig. 1. Transcriptional loss during

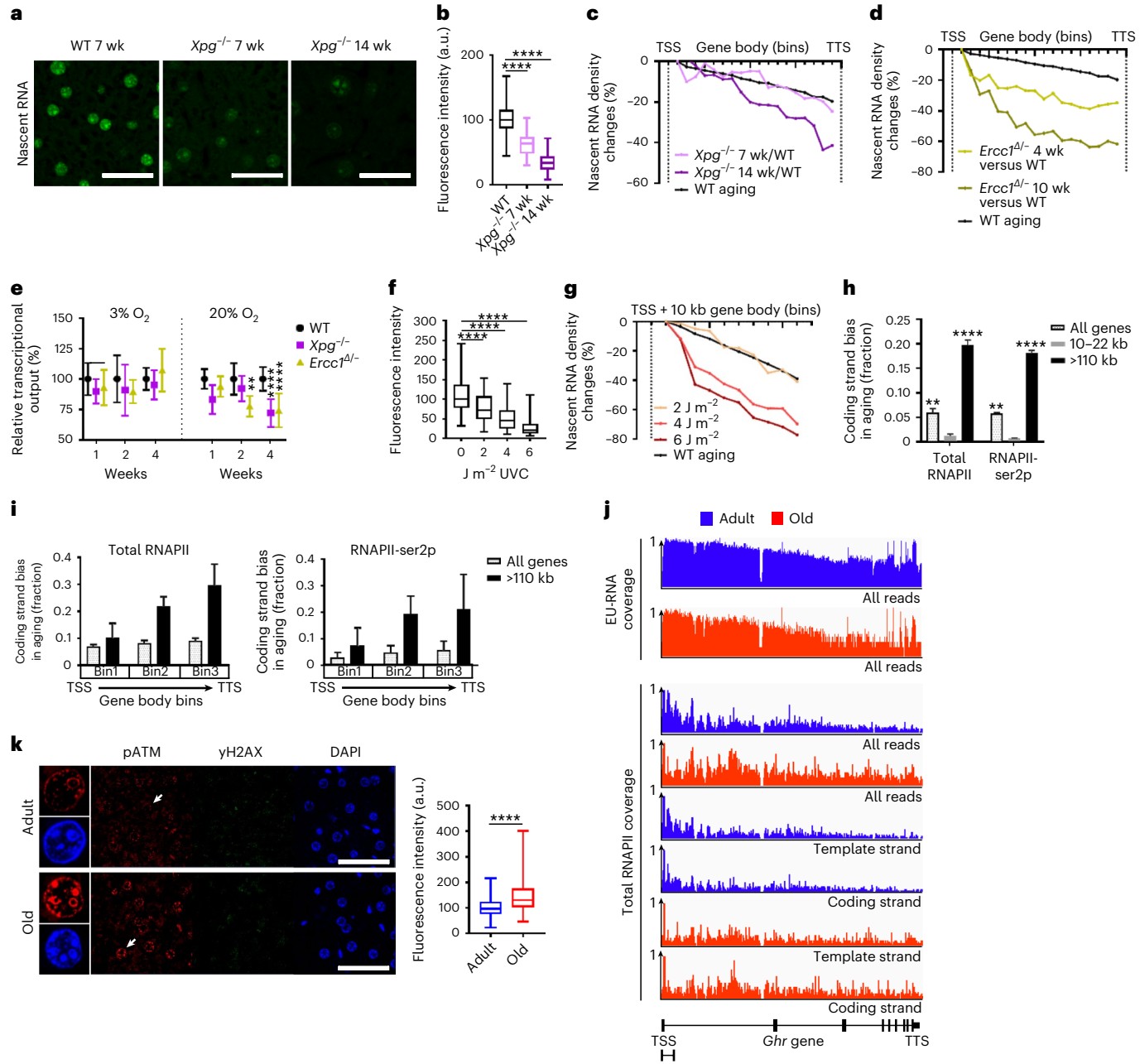

**Fig. 4 | Age-related RNAPII stalling on DNA damage. a**, EU-labeled nascent RNA (green) in liver nuclei from 7- and 14-week-old $Xpg^{-/-}$ mice compared to 7-week-old WT mice. **b**, Box and whisker plot quantification of Fig. 4a. The center lines show the medians, the box limits mark the IQR, and the whiskers indicate the minimum and maximum values. $P$ values: 7-week-old $Xpg^{-/-}$ versus WT $P = 2.4688 \times 10^{-285}$; 14-week-old $Xpg^{-/-}$ versus WT $P = 0$; two-sided unpaired $t$-test, 3 mice per group; counted nuclei $n = 916$, 864 and 738 for WT, $Xpg^{-/-}$ aged 7 and 14 weeks. **c**,**d**, Percentage EU-seq read density changes between TSS and TTS in $Xpg^{-/-}$ (**c**) and $Ercc1^{\Delta/-}$ mice (**d**) compared to WT liver aging (104 weeks, black line). **e**, Percentage decline in nascent RNA production in $Xpg^{-/-}$, $Ercc1^{\Delta/-}$ and WT quiescent MDFs after 1, 2 and 4 weeks of culturing under hypoxic (3%) and normoxic (20%) conditions. Data are the mean ± s.d. $P$ values (two-sided unpaired $t$-test) are: week 2: $Ercc1^{\Delta/-}$ versus WT: $P = 0.002353336$; week 4, $Xpg^{-/-}$ versus WT: $P = 6.13324 \times 10^{-9}$; $Ercc1^{\Delta/-}$ versus WT: $P = 1.21727 \times 10^{-9}$. Number of nuclei: 3% $O_2$, week 1: 14 $Xpg^{-/-}$ and 14 WT; 17 $Ercc1^{\Delta/-}$ and 14 WT; week 2: 15 $Xpg^{-/-}$ and 16 WT; 17 $Ercc1^{\Delta/-}$ and 13 WT; week 4: 29 $Xpg^{-/-}$ and 27 WT; 34 $Ercc1^{\Delta/-}$ and 28 WT. **f**, Box and whisker plot of fluorescent EU-labeled nascent RNA in $Ercc1^{\Delta/-}$ MDFs 24 h after UVC irradiation. The center lines show the medians, the box limits mark the IQR, and the whiskers indicate the minimum and maximum values. $P$ values (two-sided unpaired $t$-test): 2 J m$^{-2}$ versus 0 J m$^{-2}$ = $5.66445 \times 10^{-8}$; 4 J m$^{-2}$ versus

0 J m$^{-2}$ = $2.92531 \times 10^{-28}$; 6 J m$^{-2}$ versus 0 J m$^{-2}$ = $5.59594 \times 10^{-52}$. Counted nuclei: 0 J m$^{-2}$, $n = 146$; 2 J m$^{-2}$, $n = 118$; 4 J m$^{-2}$, $n = 132$; 0 J m$^{-2}$, $n = 137$. **g**, Percentage of EU-seq read densities of genes >110 kb from the TSS to 10 kb upstream in $Ercc1^{\Delta/-}$ MDFs 24 h after UVC irradiation compared to nonirradiated cells. Black line: >110 kb gene class from normal liver aging data. **h**, Bias (fraction) of sequencing reads mapping to the coding strand during WT aging from total RNAPII and RNAPII-ser2p ChIP–seq data across all genes ($n = 3,809$), short (10–22 kb, $n = 512$) and longest genes (>110 kb, $n = 779$). $P < 0.0001$, two-sided unpaired $t$-test compared to genes with gene length 1–10 kb, 3 mice per group. Data are the mean ± s.e.m. **i**, Bias (fraction) of sequencing reads mapping to the coding strand during WT aging from total RNAPII and RNAPII-ser2p ChIP–seq data through gene body (3 bins) in all genes and the longest genes (>110 kb, $n = 779$). Data are the mean ± s.e.m. **j**, Sequencing read density profiles of the $Ghr$ gene from EU-seq, total RNAPII (all reads aggregated) and total RNAPII split in coding and template strand in WT adult (blue) and aged (red) liver. **k**, Phosphorylated ATM (red) and γH2A.X (green) in adult and aged mouse liver. Right, Fluorescence intensities shown as box and whisker plots. The center lines show the medians, the box limits mark the IQR, and the whiskers indicate the minimum and maximum values. $P = 7.19752 \times 10^{-27}$ (two-sided unpaired $t$-test). Counted nuclei: adult $n = 313$; old $n = 315$; $n = 3$ mice per group. Scale bar, 50 µm.

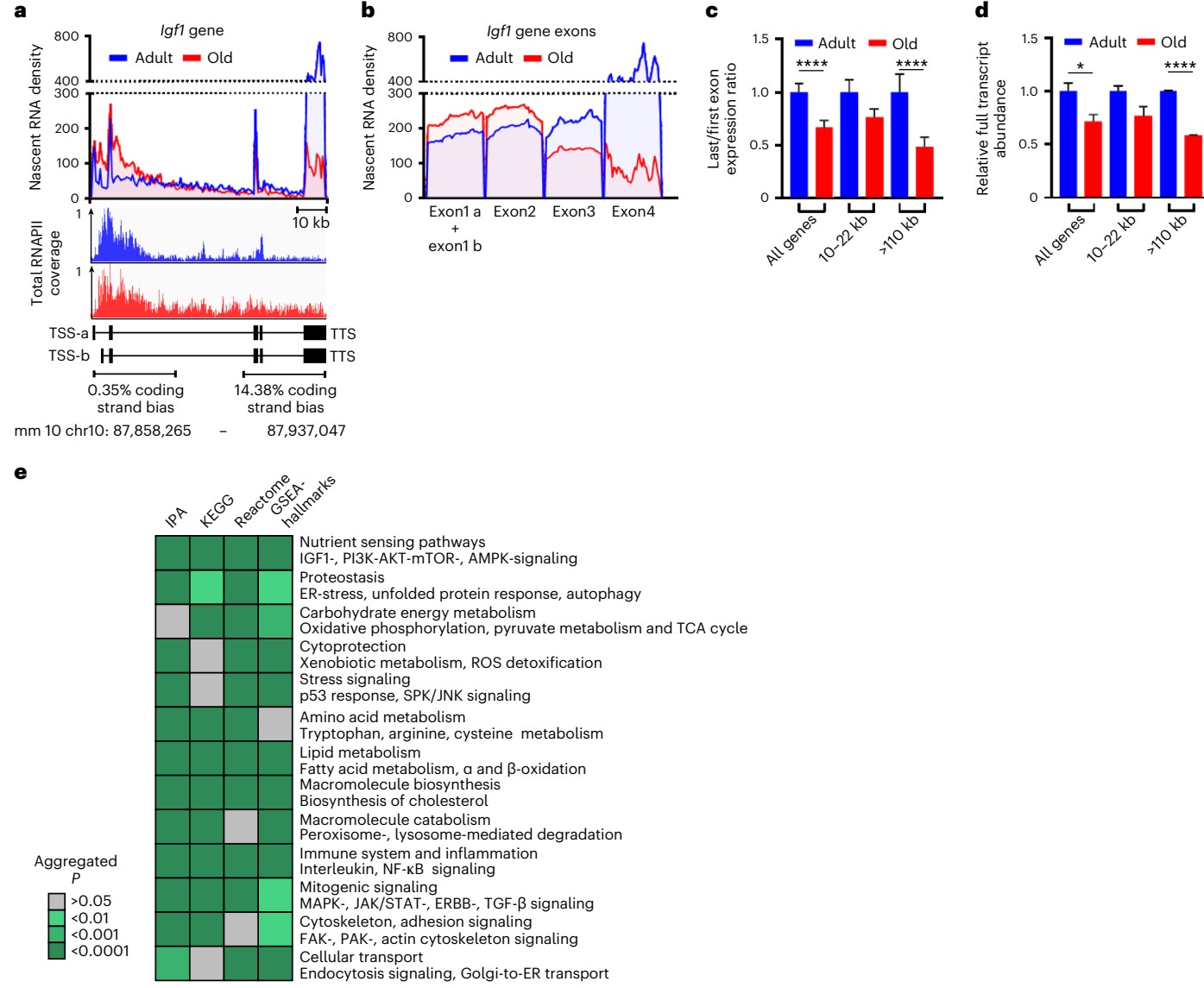

**Fig. 5 | Transcriptional stress affects mRNA output and aging-related pathways. a,b,** Sequencing density profiles of the entire *Igf1* gene (mm10, chr10:87,858,265–87,937,047), including RNAPII coding strand bias (**a**) and exons only (**b**) from EU-seq of adult (blue) and aged (red) WT mouse liver. Exons 1a and 1b are alternative start sites. **c,** EU-seq density ratios between last and first exons for all expressed genes ($P = 5.06731 \times 10^{-9}$), short (10–22 kb) and long genes (>110 kb, $P = 0.000482946$). Data are the mean ± s.e.m. $P$ values are from a two-sided unpaired $t$-test (old versus adult). **d,** Full transcript abundances (relative to adult) estimated by reads covering 3'UTR from EU-seq of all expressed genes ($P = 0.048761825$), short (10–22 kb) and long genes (>110 kb, $P = 1.78654 \times 10^{-6}$). Data are the mean ± s.e.m., $P$ values are from a two-sided unpaired $t$-test. **e,** Significant overrepresented pathways in TS[high] genes by IPA, KEGG, Reactome and GSEA-hallmarks classified by main process category (bold). Aggregated $P$ values were obtained from a Fisher's exact test. See Supplementary Table 2 for detailed pathway information.

elongation provides an explanation for reduced transcription, which, paradoxically, concurs with increasing levels of elongating RNAPII.

Previously, we reported the preferential loss of long gene mRNA expression in aged rodent liver and human hippocampus[29], later also noted in fruit fly photoreceptors[30] and brain aging[31,32]. Therefore, we tested whether gene length is implicated in GLPT. We first selected the genes from Fig. 3b with the largest age-related transcriptional decline. These GLPT[high] genes ($n = 914$) were indeed on average significantly longer compared to all expressed genes or transcriptionally upregulated or downregulated genes (Extended Data Fig. 3d). Next, we grouped all expressed genes in six gene-length classes, each containing a similar number of genes, and determined the percentage nascent RNA and RNAPII change across the gene body in aging. This analysis revealed clear gene-length-dependent opposite trends: declining transcription and increasing RNAPII occupancy (Fig. 3e,f).

Interestingly, when we plotted the mean gene length of each gene class against the percentage transcriptional decrease over the gene bodies, all classes exhibited a similar linear transcriptional regression (Fig. 3g), averaging approximately 0.35% loss per kilobase in old liver. As confirmation, the transcriptional decline in the first 20 kb from the TSS was similar across all gene lengths (Fig. 3h). Because genes >70 kb already comprised approximately 60% of the RNAPII-dependent nascent RNA pool (Extended Data Fig. 3e), long genes disproportionally contributed to reduced nascent RNA levels. The decrease in de novo RNA synthesis and increased RNAPII abundance in gene bodies entail longer residence times and lower transcriptional output of RNAPII. By quantifying the discordance between nascent RNA levels and total RNAPII occupancy (Extended Data Fig. 3f), we estimated an overall approximate 40% nonproductive RNAPII in gene bodies in 2-year-old liver in a gene-length-dependent fashion (Fig. 3i), which implies that

they are stalled. Assuming that mouse hepatocytes have a similar number of RNAPII molecules per cell as cultured human fibroblasts[33], we believe that the average 2-year-old mouse hepatocyte contains at any time >18,000 stalled RNAPII complexes during elongation (Extended Data Fig. 3g). In summary, liver aging is characterized by a gene-length-dependent, genome-wide loss of transcription elongation and increased RNAPII stalling.

## DNA damage causes transcription stalling in aging

Subsequently, we assessed whether various potential parameters were correlated with the degree of GLPT to identify a mechanism explaining RNAPII stalling. We did not find significant differences between GLPT^high genes and other gene categories (transcriptionally upregulated and downregulated; remainder) in nucleotide content across gene bodies, transcriptional error rate, alternative splicing, chromatin accessibility, histone modifications associated with euchromatin or DNA methylation patterns, which would point to epigenetic changes being responsible (Extended Data Figs. 4–6). These factors do not correlate with the degree of age-related GLPT, which is expected when such a factor is causally involved, and hence do not explain the observed transcriptional decline.

In view of gene-length-dependent transcriptional stalling, a plausible explanation is accumulation of transcription-blocking DNA damage because long genes have a higher probability to acquire stochastic lesions[26,29]. Therefore, we monitored de novo RNA synthesis in the livers of $Xpg^{-/-}$ mice, which display many features of widespread premature aging and a 20-week life span due to defects in the DNA repair pathways TCR and global genome nucleotide excision repair by which they are unable to remove transcription-stalling lesions[34]. Both EU staining and EU-seq at the age of 7 and 14 weeks (Fig. 4a–c) revealed an age-dependent, progressive, pan-nuclear decline in transcription. EU-seq in premature aging global genome nucleotide excision repair and TCR-defective $Ercc1^{\Delta/-}$ mice that display more severe liver aging pathology due to an additional defect in interstrand cross-link repair[35,36], exhibited already high transcription loss at 4 and even more at 10 weeks (Fig. 4d), showing that the extent of transcriptional decline, DNA repair deficiency and severity of liver pathology are correlated.

To further examine spontaneous, endogenous DNA damage as instigator of transcription decline, we cultured quiescent mouse dermal fibroblasts (MDFs) from $Xpg^{-/-}$, $Ercc1^{\Delta/-}$ and WT mice for 1, 2 and 4 weeks to allow endogenous DNA damage to accumulate. Quiescence avoids lesion dilution, which occurs when cells proliferate. Interestingly, $Xpg^{-/-}$ and $Ercc1^{\Delta/-}$ MDFs demonstrated a time-dependent decline in nascent RNA synthesis cultured at 20% oxygen (Fig. 4e). MDFs cultured at 3% oxygen did not display significantly reduced transcription, suggesting oxidative DNA damage as a cause of transcription loss. Next, we assessed the level of DNA damage inducing the same degree of transcriptional decline as observed in aged liver. Quiescent $Ercc1^{\Delta/-}$ MDFs were exposed to increasing doses of ultraviolet C (UVC) light, which induces known quantities of transcription-blocking DNA lesions[37]. EU staining and EU-seq demonstrated a dose-dependent transcriptional decline, in which 2 J m$^{-2}$ UVC, which corresponds to approximately 1.6 transcription-blocking lesions per 100 kb DNA[37], induced transcription levels 24 h after UV exposure similar to the livers of WT 2-year-old mice (Fig. 4f,g). These damage levels in combination with 0.35% transcription

reduction per kilobase also explain why RNAPI, RNAPIII and mitochondrial RNAP (mtRNAP) do not show a significant decline as their target RNA species are very small.

If a significant fraction of elongating RNAPII in aging is stalled by endogenous transcription-blocking lesions, it is expected that during the strand-specific DNA amplification step in the RNAPII ChIP–seq library protocol the lesion in the template strand that actually stalls the RNAPII will also impair DNA amplification of that strand, in contrast to the undamaged (coding) strand. This should lead to a strand amplification bias in favor of the coding strand that can be visualized by strand-specific ChIP–seq as shown for UV-induced transcription-blocking lesions[38]. First, we confirmed that UV-induced DNA damage leads to a coding strand bias in our ChIP–seq protocol, which disappeared after time for repair (Extended Data Fig. 7a). Importantly, in old livers we found an age-related gene-length-dependent coding strand bias in the total and RNAPII-ser2p ChIP–seq datasets (Fig. 4h). Regions with a high coding strand bias had both unaltered local DNA methylation status or nucleotide content (Extended Data Fig. 7b–f), indicating that polymerase-blocking perturbations are present in the isolated, purified DNA from aged livers, identifying them as damaged DNA. Moreover, the age-related coding strand bias increased toward the gene ends, especially in long genes (Fig. 4i), correlating with RNAPII stalling in gene bodies (Fig. 3) and TCR being more active at the beginning of genes[39]. An example is the growth hormone receptor (Ghr) gene, a >265-kb long gene frequently downregulated in aged livers across numerous independent studies[40], in $Xpg^{-/-}$ and $Ercc1^{\Delta/-}$ mutant mice[18,34] and in cell cultures exposed to UV light[41]. Ghr demonstrates a clear GLPT and increased RNAPII abundance across the gene body. We also noticed a 20% shift in reads toward the coding strand in aged livers (Fig. 4j), indicating that Ghr downregulation is the direct result of transcription-blocking lesions. DNA damage-induced RNAPII stalling causes noncanonical DNA damage checkpoint ATM phosphorylation in the absence of double-stranded DNA breaks[42], which we also observed in aged livers (Fig. 4k), thereby further demonstrating frequent transcriptional stress in aging. Because the extent of coding strand bias corresponds with the expected level when extrapolated from UV-treated cells[38], our data reveal that endogenous transcription-blocking lesions cause RNAPII stalling in a gene-length-dependent manner, which we designated age-related transcriptional stress.

## Biological significance of age-related transcriptional stress

To assess the functional significance of transcriptional stress, we quantified its effect at the most relevant level, mature mRNA. Transcriptional loss across gene bodies affected both introns and exons, as exemplified by the Igf1 gene (Fig. 5a,b), a key regulator of nutrient sensing implicated in health and life span determination[43,44], whose expression declines with age[45,46]. As expected, RNAPII accumulated on the 79-kb Igf1 gene body. Interestingly, the coding strand bias was not present in the first third of the gene body that was also characterized by normal or even higher nascent RNA and RNAPII levels, indicating that transcriptional decline correlates with coding strand bias. Thus, DNA damage-induced transcriptional stress and not promoter silencing is the driver of lower IGF1 expression in aged liver. To quantify the consequences of transcriptional stress on exons genome-wide, we calculated the first-to-last exon loss in nascent RNA, which was increased approximately 1.5-fold in aging

**Fig. 6 | Transcriptional stress in different species and tissues. a,** Percentage EU-seq read density changes of transcription elongation between TSS and TTS of expressed genes (5-bin distribution) in EU-seq data from WT aged mouse liver (black, this study, $n = 3$ per group, $n = 3,970$ genes), aged mouse kidney ($n = 2$ per group, $n = 2,135$ genes, 7.5 weeks versus 104 weeks, blue) and total RNA-seq of human tendon ($n = 4$ per group, $n = 773$ genes, 69.5 ± 7.3 years versus 19 ± 5.8 years; brown) and C. elegans ($n = 3$ per group, $n = 2,872$ genes, day 10 versus day 1 after young adult stage; green). Data are the mean ± s.e.m. **b,** Bar diagram of the overlap between GSEA aging datasets and TS^high,

promoter-upregulated and downregulated gene classes identified in our study. Significance and FDR were calculated by Fisher's exact test and Benjamini–Hochberg method. **c,** Gene enrichment ratio ($x$ axis) between identified gene groups and GSEA aging datasets in three species: mouse (top), rat (middle) and human (bottom); TS^high (left), promoter-downregulated (middle) and promoter-upregulated (right). Dot size represents the number of GEO aging datasets. If >1 dataset of a tissue was present, the mean ± s.d. and aggregated $P$ value (Fisher's exact test) are shown.

across all expressed genes and was gene-length-dependent (Fig. 5c). This was consistent with lower mRNA production in aging (Fig. 5d), providing a mechanism for previously observed decreased cellular mRNA content during aging[2,3]. This implies declining transcriptional output and skewing of gene expression toward small genes during aging.

Since transcriptional stress reduces and skews transcriptional output, we analyzed which cellular processes and pathways were most susceptible. We selected genes with a >1.5-fold first-to-last exon transcriptional loss in aging (n = 830), representing genes with high transcriptional stress levels (TS[high]), for functional examination.

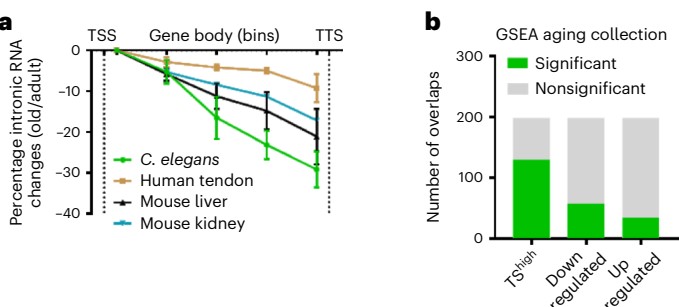

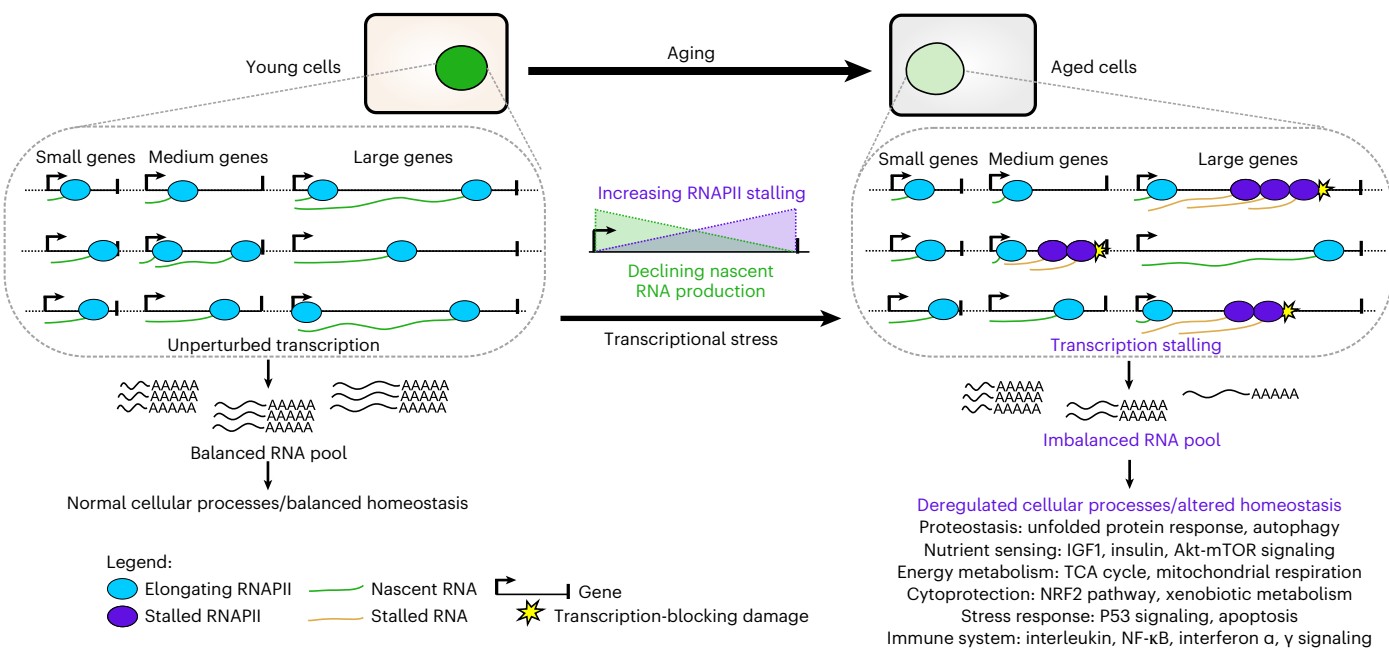

**Fig. 7 | Age-related transcriptional stress model.** Model describing RNAPII stalling by DNA damage and its consequences in aging.

Notably, we found a highly significant overlap with the overall profiles of six independent studies representing downregulated mRNAs after UVC-induced DNA damage (Supplementary Table 1), further supporting the link between transcription-blocking DNA lesions and age-related transcriptional stress. Functional examination identified several significantly overrepresented cellular pathways previously classified as hallmarks of aging[1] (Fig. 5e and Supplementary Table 2), such as the nutrient sensing pathways IGF1, insulin, growth hormone and mTOR signaling, which are all known to influence life span[1,44]. Autophagy, the unfolded protein response and the endoplasmic reticulum stress pathway were also identified, linking transcriptional stress to loss of proteostasis. Furthermore, we found key energy metabolic processes such as oxidative phosphorylation and pyruvate metabolism, which were functionally reduced by transcriptional stress in the livers of $Ercc1^{\Delta/-}$ mice[26]. Additional identified processes included immune factors, fatty acid metabolism and the NRF2 antioxidant pathway, which are all causally involved in life span and/or age-related diseases[47–50]. In conclusion, transcriptional stress appears to be a critical cause of deregulation of aging hallmark pathways and processes in WT aging mice.

**Transcriptional stress is a widespread aging phenotype**

Finally, we addressed whether transcriptional stress was confined to liver or also occurs in other organs and species. The promoter-upregulated gene set contained a B cell signature, which indicates age-related B cell infiltration[8] that also displayed transcriptional stress (Fig. 2f). EU-seq of 2-year-old mouse kidneys also showed similar GLPT as aged mouse liver (Fig. 6a). Next, we searched for and reanalyzed public total RNA sequencing aging datasets that contained sufficient reads mapping to introns representing nascent RNA. In two suitable and extensive datasets, aged human tendon[51] and *Caenorhabditis elegans*[52], we discovered a similar GLPT as in WT aged mouse liver (Fig. 6a). Since most public datasets are derived from mRNA sequencing, we also used our TS[high] gene set to match all selected age-related gene sets ($n$ = 198) in the aging perturbation library using the Enrichr tool for GSEA. For comparison, transcriptionally upregulated and downregulated gene sets were also included. We found a significant presence of TS[high] genes in 65% of all aging datasets (Fig. 6b). The transcriptionally upregulated or downregulated signatures scored much lower (Fig. 6b), while

no overlap was found with six similarly sized random gene sets, indicating that transcriptional stress is a prime driver of transcriptional changes across aging organs.

Next, we visualized which organs and tissues were significantly enriched. For organs that have multiple entries in the database, we calculated the average overlap and false discovery rate (FDR)-corrected aggregated $P$ values. As expected, age-related liver mRNA profiles from mouse and rat shared the highest similarity to the TS[high] gene set (Fig. 6c). In fact, transcriptional stress was a more dominant mechanism shaping the liver aging transcriptome than transcription regulation by promoter activity. In addition, many other organs such as kidney, heart, adipose tissue, retina, muscle, lung, neocortex and spinal cord also appeared significantly enriched for genes prone to RNAPII stalling, revealing that many organs exhibit age-related transcriptional stress, which explains overlapping gene expression patterns and also has a greater impact on gene expression than age-related promoter regulation. Not all organs displayed an mRNA transcriptional stress signature. This could be due to our transcriptional stress query gene list being biased toward liver-specific genes and/or that some organs are less prone to transcription stalling; the latter is in agreement with the segmental nature of the premature aging phenotype in TCR syndromes and corresponding mutant mice[34,36]. Proliferative tissues, for example, hematopoietic stem cells, skin and intestine appeared less vulnerable, which can be explained by the ability of DNA replication to resolve DNA damage-stalled RNAPII, which shields lesions from repair by other mechanisms[20]. Moreover, cell division dilutes DNA damage and may also enable repair. Thus, we identified transcriptional stress as a main factor shaping age-related transcriptomes and as general aging phenotype across many tissues and species.

**Discussion**

This study provides evidence that transcription-blocking DNA damage during normal aging causes frequent genome-wide elongating RNAPII stalling, which leads to reduced, gene-length-dependent transcriptional output resulting in dysregulation of many pathways known to affect aging (Fig. 7). Based on transcription-stalling similarities in UV-treated cells[38], we estimate that an initial RNAPII stalled on a lesion will block approximately three subsequent RNAPII complexes causing

queuing. Underlying mechanisms responsible for age-related gene expression changes have been largely elusive and often thought to result from active regulatory mechanisms such as promoter regulation[8], also in DNA repair mutant premature aging mouse models[17]. However, we suggest that passive transcriptional stress by DNA damage in combination with gene architecture, that is, gene length, accounts for a substantial fraction of these changes.

Each cell may suffer up to 100,000 DNA lesions per day[53], of which most are quickly repaired by dedicated repair processes. DNA damage as cause of aging is to a large extent based on premature aging syndromes with underlying genome instability, such as TCR-defective Cockayne syndrome and trichothiodystrophy, which exhibit short life span and many premature aging features predominantly in postmitotic tissues. Corresponding mouse models[34,35,54] display mRNA transcriptomes that significantly overlap with aged WT mice[18,19]. We now show that passive transcriptional stress, instead of active gene regulation, is responsible in shaping the aging transcriptome. Although we did not rule out all putative reasons explaining the observed transcriptional stress phenotype, we identified a bias toward the nontranscribed strand in RNAPII ChIP–seq data from WT aged livers, indicating transcription-blocking DNA damage in template strands as the most likely cause for RNAPII stalling. Although DNA damage accumulates during aging, it was hitherto unclear whether these levels are sufficient to elicit aging responses in WT organisms. Candidate endogenous transcription-blocking lesions, aldehydes[21], advanced glycation end products[22] and cyclopurines[23–25], can accumulate in aged organs to sufficient levels to explain the observed transcriptional stress. Thus, spontaneous, endogenous DNA damage accumulation, similar to progeroid Cockayne syndrome and trichothiodystrophy, causes transcriptional stress in normal aging.

Our data indicate how DNA damage causes aging via transcriptional stress. Transcriptional stress largely determines aging expression profiles in multiple organs impacting organ function and particularly causes the dysfunction of many aging hallmark pathways. Additionally, the stochastic nature of DNA lesions may explain transcriptional noise, which increases in aging[4–6]. Transcriptional stress could further impact cellular functioning by promoting loss of protein complex stoichiometry, a phenotype seen in aged killifish[55] and *C. elegans*[56]. Also, imbalanced expression of large and small genes due to transcription-blocking lesions in cell cultures can induce cell death and has been proposed to be a premature aging signal in Cockayne syndrome[57]. Finally, RNAPII stalling itself is also a direct cue for aging. Genetic dissection of TCR and corresponding hereditary syndromes indicates that molecular consequences of TCR mutations correlate with the severity of premature aging[20]. TCR defects that permanently stall RNAPII on DNA lesions lead to more severe forms of accelerated aging than repair defects that still permit accessibility of the lesion for other repair pathways[20]. This was further proven in mutant mice with a point mutation in RNAPII that abolishes the DNA damage-induced ubiquitination required to remove stalled RNAPII from a DNA lesion, which exhibited reduced life span and premature aging[38]. RNAPII stalled on a DNA lesion leads to R-loop formation and activation of DNA damage checkpoint ATM[42], which can induce cell death or senescence[58], thereby providing a mechanism for how RNAPII stalling leads to aging. R-loops increase in the eyes of aged fruit flies, predominantly in long genes[59], providing further proof for this scenario. We show that approximately 40% of all elongating RNAPII complexes are stalled by DNA damage in WT aged livers; thus, the identical aging signal that causes progeroid syndromes also occurs in normal aging. Interestingly, brains from patients with Alzheimer's disease also displayed reduced expression of long genes compared to age-matched controls[60], suggesting that the magnitude of transcriptional stress is involved in age-related disease etiology. Conversely, longevity-promoting intervention dietary restriction restores the loss of long gene expression[29], indicating that longevity interventions can alleviate transcriptional stress. In conclusion, we propose that endogenous DNA lesion accumulation with age triggers transcriptional stress

that shapes age-related gene expression profiles in many organs and tissues, is present over wide evolutionary distances and can explain how accumulating DNA damage causes functional decline, thereby strengthening the primary role for DNA damage in the aging process[16,17].

## Online content

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

## Methods

### Mice

Mouse housing and experiments were performed according to the Animal Welfare Act of the Dutch government, following the Guide for the Care and Use of Laboratory Animals and with the guidelines approved by the Dutch Ethical Committee in full accordance with European legislation. The institutional ethical committee for animal care and usage approved all animal protocols. WT male mice (*Mus musculus*) in F1 C57BL6J/FVB (1:1) hybrid background, were euthanized at 15 weeks and 104 weeks of age. WT C57BL6J and FVB strains are frequently obtained from the Jackson Laboratories to maintain a standard genetic background. DNA repair-deficient premature aging mouse models that were generated in house and WT littermates in F1 C57BL6J/FVB (1:1) hybrid background were euthanized at 4 and 10 weeks for *Ercc1*[Δ/−] mutants[35]; and 7 and 14 weeks for *Xpg*[−/−] mutants[34]. All animals were bred and maintained on AIN93G synthetic pellets (Research Diet Services; gross energy content 4.9 kcal g$^{-1}$ dry mass, digestible energy 3.97 kcal g$^{-1}$). Animals were maintained in a controlled environment (20–22 °C, 12 h light:12 h dark cycle) and were individually housed in individual ventilated cages under specific pathogen-free conditions at the Animal Resource Center (Erasmus University Medical Center). No statistical methods were used to predetermine sample sizes but our sample sizes are similar to those reported in previous publications[8,18,26,29,38,41]. Data collection and analysis were not performed blind to the conditions of the experiments and randomization of animals to experimental groups was not applicable. No animals were excluded from the experimental groups in any analysis.

### Cell culture

To assess endogenous DNA damage-induced de novo RNA synthesis, *Ercc1*[Δ/−], *Xpg*[−/−] and WT (all in C57BL6J/FVB F1 genetic background) MDFs were isolated from the tails of the corresponding mouse models and cultured in DMEM supplemented with 10% FCS and 1% PS at 5% $CO_2$ and 3% $O_2$.

### Nascent RNA labeling in vivo

Mice were injected intraperitoneally with 5-EU (AXXORA) 0.088 mg per gram of body weight. Five hours after intraperitoneal injection, mice were euthanized. Tissue samples were formalin-fixed for fluorescence staining or snap-frozen for the RNA isolation and ChIP experiments.

### Immunofluorescence staining

Slices measuring 3–5 µm were cut from paraffin-embedded, formalin-fixed liver pieces. Slices were mounted on microscope slides (Superfrost Ultra Plus Adhesion Slides, Thermo Fisher Scientific). For RNAPII staining, samples were deparaffinized with xylene, rehydrated with an alcohol gradient and washed with Milli-Q water before antigen retrieval (30 min in citrate buffer, pH 6). The antibodies used were: Alexa Fluor 594-RPB1 antibody (in 1:250 dilution), recognizing all forms of RNAPII independently of the phosphorylation status of their CTD (cat. no. 664908, BioLegend); RNAPII-ser2p (cat. no. ab5095, Abcam); RNAPII-ser5p (cat. no. ab5131, Abcam); phospho-ATM (Ser1981, cat. no. 4526, Cell Signaling Technology); and phospho-histone H2A.X (Ser139, cat. no. 9718; Cell Signaling Technology) all in 1:500 dilution. To reduce the background fluorescence level, a mouse-on-mouse detection kit was used (cat. no. BMK-2202, Vector Laboratories). Sections were counterstained using DAPI or Hoechst 33342.

### EU-labeled nascent RNA staining

After the xylene-based paraffin removal and rehydration steps (the antigen retrieval step was omitted for EU staining) we used the Click-iT RNA Alexa Fluor 488 Imaging Kit (cat. no. c10329; Thermo Fisher Scientific) according to the standard immunofluorescence protocol. Images were taken by a ZEISS LSM 700 system. Nascent RNA staining intensity was quantified by calculating the integrated density

values for each nuclear staining using the Fiji software[61]. Statistical significance was calculated from normalized fluorescence intensity values using an unpaired Student's *t*-test in Prism version 7.04 (GraphPad Software). For EU-labeled nascent RNA staining in vitro, cells were grown on coverslips. In confluent growth dishes, medium was replaced with fresh medium supplemented with 1% FCS and (when indicated) moved to 20% $O_2$ for the indicated time. Medium was renewed twice a week. To measure de novo RNA synthesis after UVC treatment, *Ercc1*[Δ/−] and WT MDFs (both C57BL6J/FVB F1 hybrid background) were cultured to confluency and maintained as described above followed by UVC irradiation (0, 2, 4 and 6 J m$^{-2}$ UVC) using a 254-nm germicidal lamp (Philips). The assays were performed 24 h later to allow the MDFs to recover from the immediate transcriptional effects in *trans*. To assess their transcriptional level, 1 mM EU was added to the medium for 1 h before cell collection for total RNA extraction or fixed for fluorescence staining. Cells were washed with ice-cold tris-buffered saline (TBS) and fixed for 20 min on ice in 4% formalin. Subsequently, cells were washed in 3% bovine serum albumin (BSA) in TBS and permeabilized using 0.5% Triton X-100 in TBS for 20 min at room temperature. The coverslips were then washed twice with 3% BSA in TBS and incubated with Click-iT reaction mix (Invitrogen) for 30 min. After the Click-iT reaction, cells were washed once with 3% BSA in TBS and once with TBS before being incubated in TBS containing 1:1,000 Hoechst 33342 for 30 min. Samples were mounted using Prolong Diamond (Invitrogen). Images were obtained with a LSM700 ZEISS Microscope and EU staining intensity in nuclei was quantified with Fiji (Image J 1.53q).

### Total RNA-seq and EU-seq library generation and sequencing

Total RNA was isolated from snap-frozen liver and kidney slices or scraped cells using the miRNeasy kit (QIAGEN) including the on-column DNase step (RNase-Free DNase Set, QIAGEN). RNA quality and quantity were estimated with the Bioanalyzer (Agilent Technologies) and only high-quality RNA (RNA integrity number >8) was used for further analyses. Total RNA sequencing was performed as described elsewhere[62]. To selectively isolate EU-labeled nascent RNA, we used the Click-iT nascent RNA Capture Kit (cat. no. c10365, Thermo Fisher Scientific): biotin azide was attached to the ethylene groups of the EU-labeled RNA using Click-iT chemistry. The EU-labeled nascent RNA was purified using MyOne Streptavidin T1 magnetic beads. Captured EU-RNA attached on streptavidin beads was immediately subjected to on-bead sequencing library generation using the TruSeq mRNA Sample Preparation Kit v2 (Illumina) according to the manufacturer's protocols with modifications. The first steps of the protocol were skipped; directly on-bead complementary DNA (cDNA) was synthesized by reverse transcriptase (Super-Script II) using random hexamer primers. The cDNA fragments were then blunt-ended through an end-repair reaction, followed by dA-tailing. Subsequently, specific double-stranded barcoded adapters were ligated and library amplification for 15 cycles was performed. PCR libraries were cleaned up, measured on an Agilent Bioanalyzer using the DNA1000 assay, pooled at equal concentrations and sequenced per three in one lane on a HiSeq 2500.

### ChIP–seq library generation and sequencing

Snap-frozen liver was minced in ice-cold PBS, homogenized in a Dounce homogenizer and filtered through a cell strainer (Falcon). After adding formaldehyde (total 1%), the homogenate was shaken on ice for 10 min and quenched with glycine. Pelleted homogenate was washed with ice-cold PBS, resuspended in cell lysis buffer (0.25% Triton X-100, 10 mM EDTA, 0.5 mM EGTA, 20 mM HEPES, pH 8.0, cOmplete EDTA-free and PhosSTOP, Sigma-Aldrich) and incubated for 10 min on ice. Samples were centrifuged and resuspended in nuclei lysis buffer (0.15 M NaCl, 1 mM EDTA, 20 mM HEPES, pH 8.0, cOmplete

EDTA-free and PhosSTOP). Samples were further homogenized by Dounce homogenizer and incubated on ice for 10 min. The nuclear fraction was resuspended in sonication buffer (50 mM HEPES, pH 7.8, 140 mM NaCl, 1 mM EDTA, 1% Triton X-100, 0.1% sodium deoxycholate, 1% sodium dodecyl sulfate, cOmplete EDTA-free and PhosSTOP). The chromatin was sonicated with a Bioruptor (Diagenode) sonicator into 100–500 bp fragments and centrifuged to remove any remaining cell debris. From the supernatant, 15 µg chromatin was used for one round of immunoprecipitation. For ChIP–seq, five samples were pooled. Dynabeads M-280 Sheep Anti-Rabbit IgG beads were used for the immunoprecipitation step. Chromatin samples were precleared with beads at 4 °C, for 2 h. The precleared chromatin samples were rotated overnight at 4 °C with the RNAPII antibodies: RNAPII-ser2p, RNAPII-ser5p or RNAPII RPB1-NTD-specific antibody (clone D8L4Y, Cell Signaling Technology). Dynabeads were added for 2 h to the samples to pull down protein–DNA complexes. After immunoprecipitation, samples were washed twice with the following buffers: sonication buffer, twice with buffer A (50 mM HEPES, pH 7.8, 500 mM NaCl, 1 mM EDTA, 1% Triton X-100, 0.1% sodium deoxycholate, 0.1% SDS, cOmplete EDTA-free and PhosSTOP), twice with buffer B (20 mM Tris, pH 8, 1 mM EDTA, 250 mM LiCl, 0.5% NP-40, 0.5% sodium deoxycholate, cOmplete EDTA-free and PhosSTOP) and finally twice with Tris-EDTA buffer (10 mM Tris, pH 8, 1 mM EDTA). The bound fraction of the chromatin was isolated using the IPURE DNA recovery for ChIP Kit (Diagenode). Sequencing libraries were generated using the Illumina TruSeq ChIP Library Preparation Kit. Samples were sequenced on the HiSeq 4000 platform.

## Sequence read mapping

EU-seq reads were preprocessed with the quality control software FastQC v.0.11.9, FastQScreen v.0.14.0 and Trimmomatic v.0.35 (ref. [63]) using the parameters: SLIDINGWINDOW:4:15 LEADING:3 TRAILING:3 ILLUMINACLIP:adapter.fa:2:30:10 LEADING:3 TRAILING:3 MINLEN:36. The remaining reads were successively aligned to the mouse ribosomal DNA (BK000964.3), mitochondrial sequences (UCSC, mm10) and mouse reference genome (GRCm38/mm10) using Tophat2 v.2.0.9 (ref. [64]) with default settings except for the -g 1 option. ChIP–seq reads were aligned to the mm10 mouse reference genome using Bowtie[65] v.2.1.0. The public total RNA-seq dataset applied the same mapping algorithm with EU-seq using the corresponding reference genome (hg19 and ce10) to study the nascent RNA dynamics in aging among species. The same mapping algorithm as used with ChIP–seq was applied to the other public data.

## Definition of unique intron, exon, gene regions and gene groups

All RefSeq (release: 95) genes, exons and introns were extracted from the UCSC Genome Browser[66] and the gene lists were collapsed to the longest transcript for each gene. Genes with regions overlapping another coding or noncoding gene were removed. Thus, genes having only regions unique to a specific RefSeq gene were used for further analysis. In some experiments as indicated in this study, specific genomic regions (from TSS to 1 kb downstream; intronic regions only; from TSS to 20 kb downstream; 3′UTR; first and last exon of expressed genes; around TSS region (−0.75 kb to +0.75 kb and −0.3 kb to +0.3 kb) were generated in the same manner. To investigate the productive elongation process per gene, genomic regions around the TSS and TTS of genes were divided into $k$ proportional bins ($k = 20$ by default; due to the data quality in different datasets, the number of bins varies (details following)). Genes with length smaller than 10 kb were removed from the study to avoid too many reads mapping to short genes overlapping bins in the gene proportional elongation analysis.

A Python pipeline (K_bining.py) was created that takes aligned RNA/EU/ChIP–seq reads in BAM/BW format as input to quantify reads in the transcription elongation region of genes and HTSeq was performed for read quantification in the aforementioned genomic regions. Reads per million (RPM) was applied to normalize different sequencing libraries to exclude technical variation (especially sequencing depth) in further studies. The 'all genes' gene set comprises all genes with at least one read mapping in the first kilobase. A gene set with genes that have at least 1 RPM in each of the 20 bins was constructed and termed 'all expressed genes'. To study read distribution across gene bodies and due to sequence depth and data quality, different number of bins ($k$) and the number of all expressed genes ($n$) were selected for each EU-seq and public total RNA dataset collection. Therefore, the 'all expressed genes' set in WT aging contain: $n = 3,970$ genes and were divided into $k = 20$ bins (>90% of all EU-seq reads are mapped to these genes). $Ercc1^{\Delta/-}$ mice ($k = 20$, $n = 2,430$); $Xpg^{-/-}$ mice ($k = 20$, $n = 3,842$); UV-treated $Ercc1^{\Delta/-}$ MDFs ($k = 10$, $n = 1974$); WT aging kidney ($k = 20$, $n = 2,135$); human tendon ($k = 5$, $n = 773$); C. elegans ($k = 5$, $n = 2,872$). To match WT aging EU-seq data with the corresponding RNAPII ChIP–seq data (generated from the same liver), the corresponding genes from the 'all expressed genes' gene set were also selected in the RNAPII ChIP–seq datasets. The intra-sample-specific background was determined by calculating the reads in the intergenic regions and proportionally removed. The overall background signal was subtracted using the DNA input samples. To biologically define the 'all expressed genes' ($n = 3,970$) in WT aging, we performed a $k$-mean clustering analysis combined with EU-seq and total ChIP–seq reads between adult and old samples. Under the criterion describes in Extended Data Fig. 3a, we defined the four main patterns found in $k$-mean cluster analysis as four biological groups: promoter-upregulated genes, $n = 778$ (EU-seq and RNAPII ChIP–seq level increased across three bins); promoter-downregulated genes, $n = 394$ (EU-seq and RNAPII ChIP–seq level decreased across three bins); GLPT$^{high}$ genes, $n = 914$ (steep EU-seq level progressive decrease, steep RNAPII ChIP–seq level increase across three bins); remainder genes, $n = 1,884$ (mild EU-seq level progressive decrease, mild RNAPII ChIP–seq increase across three bins). To study the relationship between gene length and transcriptional stress phenotype, the expressed genes ($n = 3,970$) in WT mice were divided into six groups according to their length, each containing a similar number of genes: 10–22 kb ($n = 662$, average = 16.47 kb, median = 16.75 kb); 22–30 kb ($n = 644$, average = 26.87 kb, median = 26.94 kb); 30–50 kb ($n = 788$, average = 40.19 kb, median = 39.68 kb); 50–70 kb ($n = 587$, average = 59.18 kb, median = 59.02 kb); 70–110 kb (n = 643, average = 87.93 kb, median = 86.75 kb) and >110 kb ($n = 646$, average = 199.47 kb, median = 160 kb). In figures measuring gene class behavior, we first calculated the per gene the average signal from $n = 3$ mice followed by averaging the signal for all genes in the gene class.

## Gene function enrichment analysis

Gene ontology and functional clustering analyses of TS$^{high}$ genes were performed by using multiple databases and software: Ingenuity Pathway Analysis, GSEA v.4.2.2 that also includes the Kyoto Encyclopedia of Genes and Genomes (KEGG), Reactome Pathway Databases, Aging Perturbations from GEO and the down datasets from Enrichr[27,28,67,68]. Datasets in the Aging Perturbations from the GEO and the down datasets from Enrichr were only included if young/adult >8 weeks, old is >14 months and age difference between young/adult and aged organs is >6 months (mouse, rat); human old is >56 years with at least an age difference of >12 years. We adopted a threshold FDR < 0.05. Aggregated $P$ values for the main identified biological processes were calculated by combining the $P$ values of the corresponding detected subpathways using Fisher's exact test.

## Genome-wide characterization of the aging transcriptome

The RNAPII travel ratio is the ratio between RNAPII on the TSS and RNAPII in the first 1-kb gene body in old and adult. The total RNAPII density value around the TSS (±300 bp) was divided by the total RNAPII density value on the first 1-kb gene body measured from 300 bp

downstream of TSS for each gene. To compare nascent RNA synthesis from the first 1-kb region of the first introns or from the TSS to 1 kb downstream of all genes between adult and old mouse livers, all datasets were only normalized based on sequencing read depth, not correcting for the approximately 1.5-fold reduced number of intronic sequences in aged liver. To measure productive transcription elongation, all expressed genes were divided into $k$ proportional bins, where mean read counts from the EU-seq and RNAPII ChIP–seq dataset was calculated. Counts were subsequently normalized to the first bin and plotted. The percentage density changes per bin were calculated by: $old_{(readcount)} / adult_{(readcount)} \times 100\%$. Since the first and last bin includes the signal at the TSS in which RNAPII promoter proximal pausing is present, and the TTS in which RNAPII accumulates, we defined the middle 18 bins as the transcription elongation phase. The 3-bin heatmap is derived from the 18-bin data by aggregating 6 subsequent bins and calculating $log_2$ fold changes of every gene between old and adult samples for both EU-seq and total RNAPII ChIP–seq reads. To determine the percentage increase RNAPII stalling or unproductive RNAPII in aged livers (Extended Data Fig. 3f), we assumed a baseline in adult livers in which the total nascent RNA level in the elongation phase (18 bins) is the result of the total RNAPII levels in the elongation phase (18 bins). The mean relationship across $n = 3$ mice is set as the baseline. Since there is an increase in RNAPII levels and a reduction in nascent RNA levels across the gene body in aged liver, the expected number of total RNAPII ChIP–seq reads was calculated that should support these nascent RNA levels based on the adult liver baseline, that is, the ratio of EU-seq read counts and total RNAPII ChIP–seq read counts in the elongation phase (18 bins). Subsequently, the observed total RNAPII ChIP–seq read count per sample was determined in each aged liver sample. Then, we subtracted the expected total RNAPII ChIP–seq read count from the observed total RNAPII ChIP–seq read count and divided this by the expected read count in aged liver: RNAPII stalling in aging (%) = (observed RNAPII read count − expected RNAPII read count) / expected RNAPII read count × 100%. To analyze nucleotide composition, the top 50 genes from GLPT$^{high}$ and bottom 50 from the remainder genes were selected within the length range of 70–110 kb and were divided into 35 bins from TSS to TSS + 70 kb (2 kb per bin). The nucleotide composition percentage (cytidine, thymine, adenine and guanine) was determined by Qualimap v.2.21 (ref. [69]). The transcription error ratio in the EU-seq datasets was calculated using BioConductor seqTools (R v.1.2.0. and IRanges packages)[70]. Total error rates were calculated as the percentage of total reads with a mismatched base at each read position during the alignment step. Analysis of EU-seq read abundance at splicing donor and acceptor sites was carried out using a custom-written script: Splicingdonor&acceptorfinder.py in which the expression values from ±49 bp around the splicing donor and acceptor site for all selected genes were captured by the HTSeq v.0.6.0. Alternative splicing events were detected by Astalavista v.4.0 (ref. [71]) with default settings. DNA methylation status was detected by Qualimap v.2.21 and deepTools v.2.0 (ref. [72]). Average RNAPII profiles at promoters (±750 bp around the TSS) and average histone modification profiles (H3K27ac, H3K4me3 and DNA methylation) at the TSS and gene bodies were plotted using HOMER v.4.11 software (annotatePeak.pl command)[73].

To calculate the number of RNAPII stalled on a lesion compared to queuing behind the initially stalled RNAPII, we first estimated the number of DNA lesions in our expressed genes dataset, which has a total length of 280,010,046 bp. With a lesion density of 1.6 per 100,000 bp in a diploid genome, we expect approximately 8,960 DNA lesions. Since DNA lesions are equally occurring on both the coding and template strands and the latter is only important for RNAPII stalling, there are 4,480 DNA lesions in the template strand of the selected gene set. If we assume that all DNA lesions are obstructing an RNAPII complex and we have an estimated 18,000 stalled RNAPII complexes per cell, we estimate that for every RNAPII stalled on a DNA lesion three RNAPII

complexes are queuing. The strand bias analysis in the RNAPII ChIP–seq data was done as described elsewhere[38], which is based on the observation that PCR amplification of RNAPII ChIP–seq libraries is biased toward the coding strand if there is a transcription-blocking DNA lesion in the template strand on which RNAPII is stalled, with some modifications. We first monitored whether our specific ChIP–seq protocol could detect strand bias and optimized the analysis by using our previously published RNAPII ChIP–seq data after UVB treatment[74]. In short, forward and reverse reads from RNAPII ChIP–seq were separated and processed by SAMtools v.1.9 (ref. [75]) and counted by BEDtools v.2.27.1 (ref. [76]). For each gene in the selected gene set, we first corrected for the orientation of the template strand (forward/reverse strand) because genes are located on both the forward and reverse strand. Then, we calculated for each gene in the dataset the fraction bias toward the coding strand and subsequently the strand bias was calculated across all expressed genes in the gene set.

### Statistical reproducibility and modeling

In vitro experiments are based on triplicates of independent experiments and the plots are presented as the means, unless otherwise indicated. Details of the statistical tests and quantifications used in this study are described in the corresponding parts of the main text, figure legends or Methods. Data distribution was assumed to be normal but this was not formally tested. All statistical tests were performed with Prism or the packages or functions implemented in R (edgeRpackage and fisher.test functions) except for the enrichment analysis with, GSEA, Enrichr and Ingenuity Pathway Analysis, which were performed by and thoroughly described in their Web applications.

A statistical and probabilistic framework was generated for EU incorporation for a range of distances between EU molecules, and in case of a 1.5-fold reduction for a range of EU incorporation distance differences. The probability that at least one EU is incorporated into nascent RNA was modeled in the situation where there is a 1.5-fold reduction in EU incorporation in old mouse livers due to lower or slower uptake or processing. The assumptions were: (1) as there is at least a >400-fold surplus of biotin for every incorporated EU in nascent RNA in the Click-iT reaction, the reaction is saturated or follows the same asymptote; (2) only one EU incorporation per RNA molecule is sufficient to isolate that specific molecule; (3) EU incorporation is a stochastic process in which the concentration of available EU in the total nucleotide pool linearly correlates with the distance between EU molecules in the nascent RNAs. If there is an EU availability difference between adult and old mice, it is expected that in short RNA species (≤300 nucleotides) the probability of at least one EU incorporation is significantly lower and thus we would empirically observe a lower percentage sequence read mapping to such small RNA species in aged liver. The process of EU incorporation was modeled into nascent RNA species by means of a Poisson process. Specifically, one can think of the number of EU incorporations into nascent RNA as a Poisson process not in time, as it is generally used, but in length as measured in nucleotides. Mathematically, if $X(t)$ is a Poisson process then the probability that there is no event in a time interval $(0,t)$ reads $exp(-\lambda t)$ where $\lambda$ is the intensity of the Poisson process. Equally, the probability that there is at least one event in the time interval $(0,t)$ is thus $1 - exp(-\lambda t)$. For each RNA species in our specified RNA length classes identified in the EU-seq datasets, the probability that at least one EU has been incorporated was subsequently computed using the formula above. Clearly, since $1 - e^{-\frac{1}{x}} > 1 - e^{-\frac{1}{y}}$ for all $x < y$, Poisson processes with higher intensity will necessarily exhibit a larger probability that at least one EU has been incorporated than Poisson processes of lower intensity. Three groups of RNA species were examined: (1) ≤300 nucleotides (number of RNA species $n = 7,932$); (2) between 1,000 and 3,000 nucleotides (number of RNA species, $n = 1983$); and (3) between 2,000 and 4,000 nucleotides (number of species, $n = 1,802$). The number of RNA species

reflect the total number present in the *Mus musculus* genome database (Ensembl). The latter two classes, although still representing short RNA species, are incorporated as a positive control in which a difference, if there is 1.5-fold less EU available, is not expected. In all cases, the probability vectors were not Gaussian as calculated by Kolmogorov–Smirnov test; thus, for each fixed intensity of the underlying Poisson process, the median and interquartile range (IQR) for the probability that at least one EU is incorporated are calculated. Significance between 1.5-fold-apart intensities was calculated by the Mann–Whitney *U*-test.

### Reporting summary

Further information on research design is available in the Nature Portfolio Reporting Summary linked to this article.

### Data availability

EU-seq and ChIP–seq data have been deposited at the NCBI Sequence Read Archive website and are publicly available (accession no. PRJNA603447). The microscopy images reported in this paper will be shared by the lead contact upon reasonable request. Several public datasets were reanalyzed including: total RNA-seq data from the human tendon[51] (https://www.ebi.ac.uk/arrayexpress/experiments/E-MTAB-2449/) and *Caenorhabditis elegans*[52] (https://www.ncbi.nlm.nih.gov/bioproject/?term=PRJNA357503) for Fig. 6a, DNA methylation (https://www.ncbi.nlm.nih.gov/geo/query/acc.cgi?acc=GSE95361)[77], histones H3K27ac and H3K4me3 (https://www.ncbi.nlm.nih.gov/bioproject/?term=PRJNA281127) for Extended Data Figs. 5 and 6, MNase-seq (https://www.ncbi.nlm.nih.gov/geo/query/acc.cgi?acc=GSE58005)[78] and RNAPII ChIP–seq data from UVB-irradiated cells (https://www.ncbi.nlm.nih.gov/bioproject/?term=PRJNA230028) for Extended Data Fig. 7a. Source data are provided with this paper.

### Code availability

All software used in this study is published and cited either in the main text or Methods. All original code has been deposited at: https://github.com/Pothof-Lab/Transcriptional-Stress. Data analysis approaches using published software packages are described in the Methods.

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

### Acknowledgements

We thank M. Tresini, G. Garinis, P. Mastroberardino and C. Milanese for providing reagents and protocols. We thank Y. van Loon and W. Vermeij for general assistance with mouse experiments. We thank P. van Arp for RNA sequencing support. RNA sequencing was done in the ErasmusMC HuGeF facility run by A. G. Uitterlinden and J. van Meurs and the ErasmusMC Biomics facility run by W. van Ijcken. RNAPII ChIP–seq was performed by the GenomEast platform, a member of the 'France Génomique' consortium (ANR-10-INBS-0009). We acknowledge financial support from the National Institutes of Health/National Institute of Aging (P01 AG017242; DNA repair, mutations and cell aging) to J.H.J.H. and J.P., European Research Council Advanced Grant Dam2Age to J.H.J.H., the European commission EU ITN Address (GA-316390) to J.H.J.H., Dutch research organization ZonMW Memorabel project no. 733050810 to J.P. and J.H.J.H., Deutsche Forschungsgemeinschaft (German Research Foundation) project no. 73111208-SFB829 to J.H.J.H. and European Joint Programme on Rare Diseases (EJPRD TC-NER) to J.H.J.H. The funders had no role in study design, data collection and analysis, decision to publish, or preparation of the manuscript.

### Author contributions

J.P. conceived and supervised the project. J.P., A.G. and J.C. designed the experiments. J.P., A.G., J.C. and J.H.J.H. interpreted the data. J.P., A.G. and J.C. wrote the manuscript and prepared the figures. J.H.J.H. critically read, introduced the age-related loss of long genes concept and edited the manuscript. R.M.C.B. and S.B. performed the mouse experiments and EU injections. A.G. performed and analyzed the EU and RNAPII staining in WT mice. S.B. performed and analyzed EU staining in *Xpg*−/− mice. A.G. developed the EU-seq and prepared all sequencing samples. J.C., A.G. and K.W.J.D. designed the EU-seq analysis pipeline. J.C. and K.W.J.D. created the EU-seq mapping and normalization pipeline. J.C. designed the algorithms and wrote the scripts for the data analyses. J.C. and A.G. performed the bioinformatics analyses. J.J.P.G.D. performed the in vitro transcriptional stress accumulation assays. M.R. performed the statistical analyses. S.T.B. and M.P.B. performed the EU staining in UV-exposed cell cultures and analyzed cellular senescence.

### Competing interests

The authors declare no competing interests.

### Additional information

**Extended data** is available for this paper at https://doi.org/10.1038/s41588-022-01279-6.

**Correspondence and requests for materials** should be addressed to Joris Pothof.

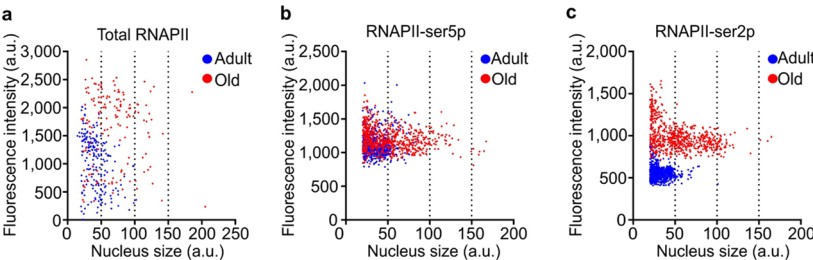

**Extended Data Fig. 1 | Relation between RNAPII fluorescent intensity and corresponding nuclear sizes measured in individual hepatocytes, related to Fig. 1. a–c**, XY-scatterplots of (**a**) total RNAPII, (**b**) RNAPII-ser5p and (**c**) RNAPII-ser2p; A.U.: arbitrary units. blue: wildtype adult livers; red= old liver. n = 3 mice/group. Total number of counted nuclei: **a**, adult: n = 206, old: n = 155. **b**, adult, n = 748; old n = 701. **c**, adult: n = 984; old, n = 674.

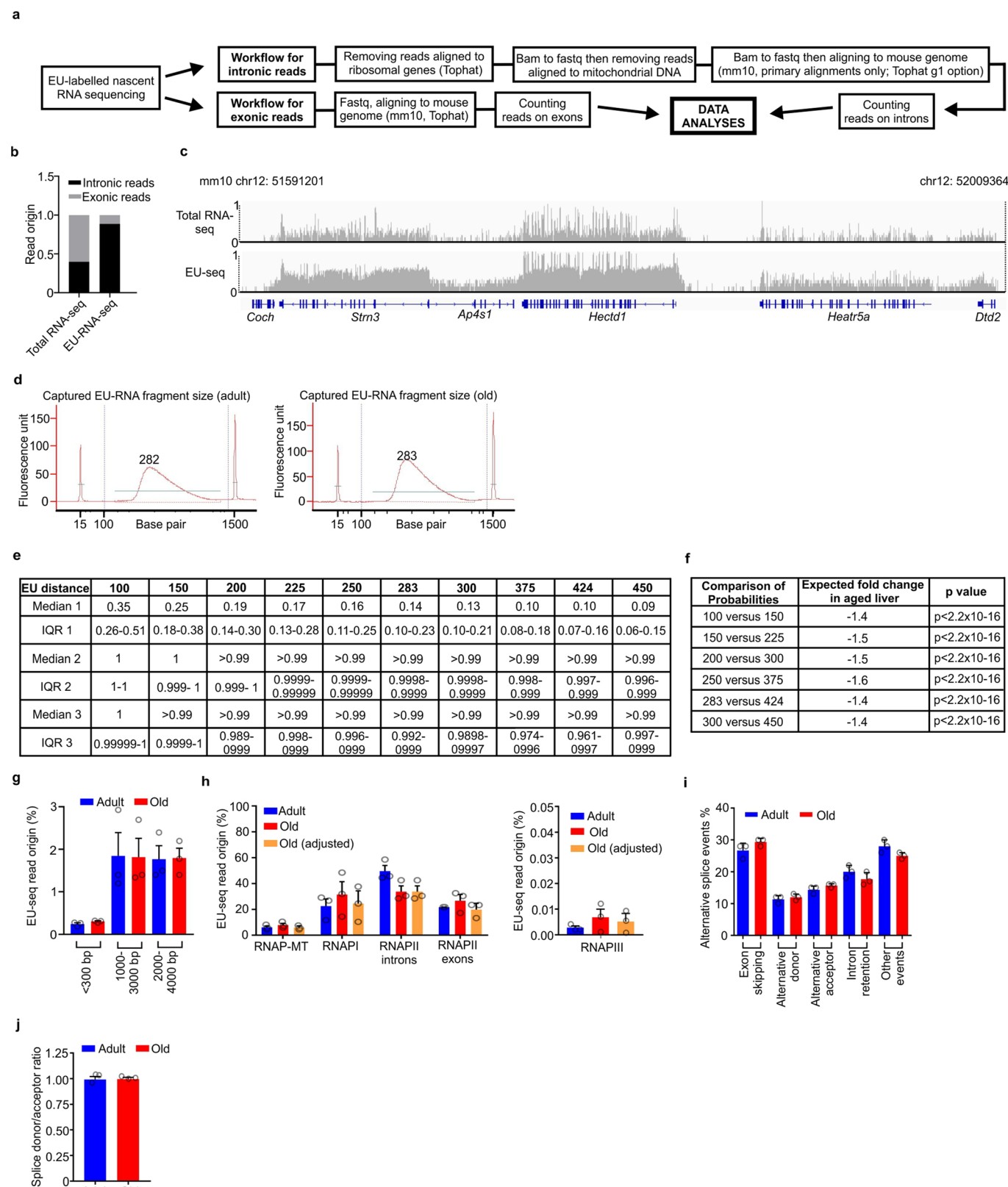

**Extended Data Fig. 2 | See next page for caption.**

**Extended Data Fig. 2 | Nascent RNA sequencing (EU-Seq) computational analysis and controls, related to Fig. 1. a**, EU-labelled nascent RNA sequencing (EU-seq) analyses flowchart. **b, c**, Sequence read distribution in EU-seq and total-RNA sequencing in introns and exons (b) and mapped on genes (c), showing increased number of intronic reads in EU-seq, indicating nascent RNA enrichment. **d**, Bioanalyzer plots of on-bead synthesized cDNA to generate EU-seq libraries. Since reverse transcriptase cannot synthesize cDNA through biotin covalently bound to DNA, cDNA is only generated between covalently EU-bound biotins or from EU-bound biotin to the RNA end. Adult and old cDNA is almost identical, indicating similar EU incorporation rates. **e**, table depicting EU incorporation for a range of distances between EU molecules as modelled by Poisson process. Three groups of RNA species were examined: 1) ≤300 nucleotides (n = 7932), 2) 1000 to 3000 nucleotides (n = 1983), and 3) 2000 to 4000 nucleotides (n = 1802). Shown is median and Interquartile Range (IQR). **f**, statistical and probabilistic framework table depicting 1.5-fold reduction for a range of EU incorporation distance differences. If EU incorporation differs between adult and old, it is expected that in RNA species ≤300 nucleotides the probability of at least one EU incorporation is significantly lower. For each specified pair of 1.5-fold apart intensities, the expected fold change in aged liver was calculated and the corresponding 7932-dimensional vectors of probabilities were compared by Mann-Whitney U-test. There is a very high, significant probability that a difference should be observed between adult and old (p < 2.2 ×10⁻¹⁶, column 3) if there is a 1.5-fold reduction in EU incorporation in aged liver. **g**, percentage sequence reads in EU-seq datasets mapping to RNA species length categories i) ≤300 nucleotides, ii) 1000 to 3000 nucleotides, and iii) 2000 to 4000 nucleotides. The ≤300 nucleotides length category shows a non-significant (p-value = 0.061093, two-sided unpaired t-test) 1.2-fold increase in aged liver, indicating that it is unlikely that EU availability is different between adult and aged livers. Data are mean ± SEM. **h**, percentage EU-labelled nascent RNA reads synthesized by different RNA polymerases (RNAPI-II-III, and mitochondrial RNAP (RNAP-MT)). Adjusted old values (orange) were calculated by proportional compensation of nascent transcription reduction as observed in Fig. 1a. n = 3 mice/group. Data are mean ± SEM. **i**, Alternative splicing in EU-seq data. n = 3 mice/group. Data are mean ± SEM. **j**, Ratios of splice donor and acceptor sites in EU-seq. n = 3 mice/group. Data are mean ± SEM.

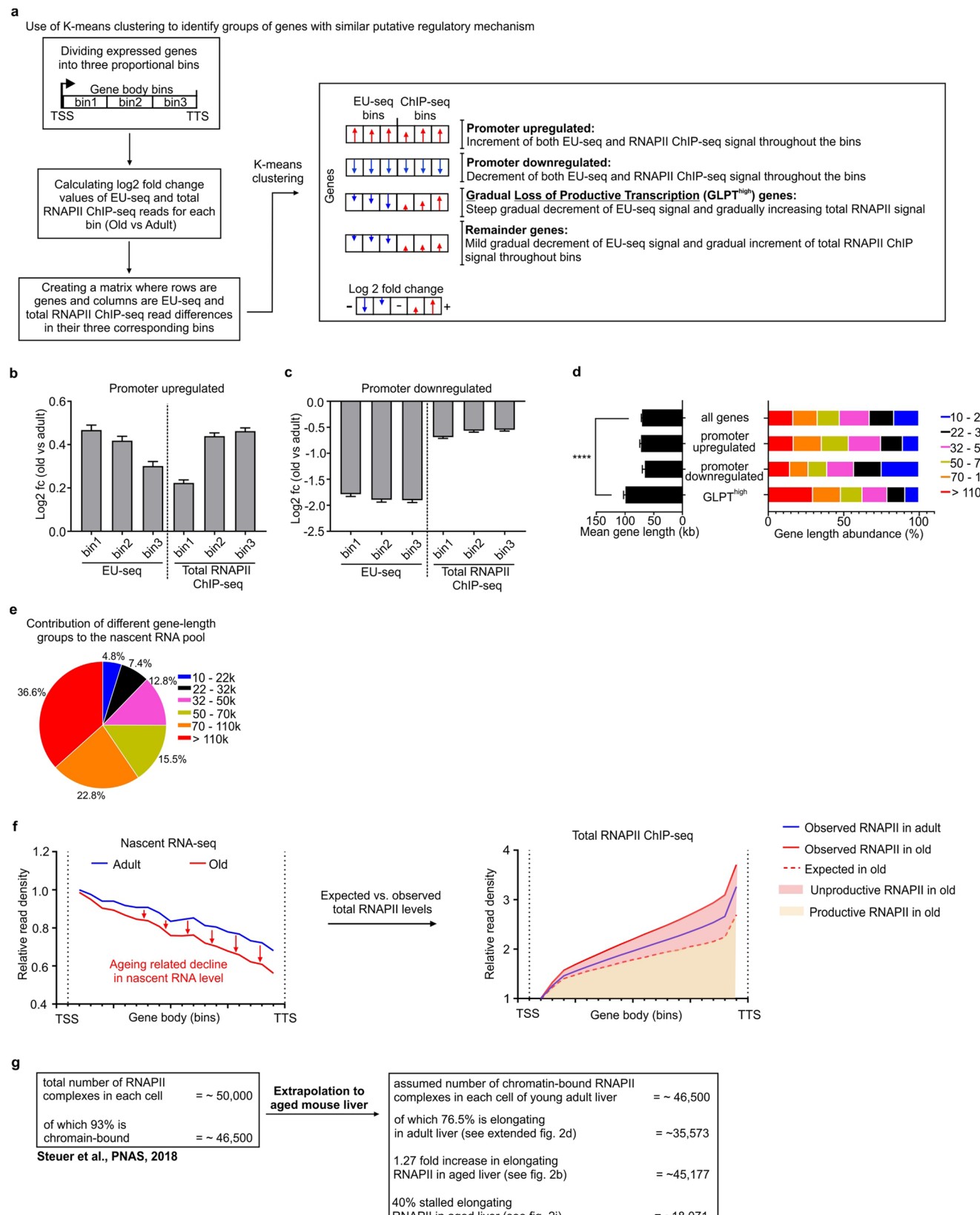

**Extended Data Fig. 3 | See next page for caption.**

**Extended Data Fig. 3 | Gene category classification, transcription characteristics of gene classes and RNAPII stalling calculation, related to Figs. 2 and 3. a**, Schematic representation of k-means clustering-based identification of putative regulatory mechanisms behind the aging-related gene expression changes. **b**, **c**, The mean Log2-fold change of EU-seq and total RNAPII ChIP-seq reads in aging liver throughout the gene bodies (3 bins) of (**b**) 'Promoter-upregulated' genes (n = 778) and (**c**) 'Promoter downregulated' genes (n = 394). Calculated from main Fig. 2 f. Data are mean ± SEM. **d**, Average gene length (left panel) of all expressed genes, promoter-upregulated genes,

promoter-downregulated genes, and genes with a high gradual loss of productive transcription (GLPT^high) as seen in Fig. 2a and classified by gene length (right panel) Groups: 10–22 kb (blue; n = 662); 22–32 kb (black; n = 644); 32–50 kb (pink; n = 788), 50–70 kb (yellow; n = 587), 70–110 kb (orange; n = 643) and >110 kb (red; n = 646). Data are mean ± SEM. P-value = $7.84163 \times 10^{-22}$, two-sided unpaired t-test. **e**, The contribution (%) of each gene-length class to the total nascent RNA pool in adult samples. **f**, Calculation of the fraction of unproductive RNAPII complexes in aged liver. **g**, Estimation of the number of stalling RNAPII complexes in aged liver.

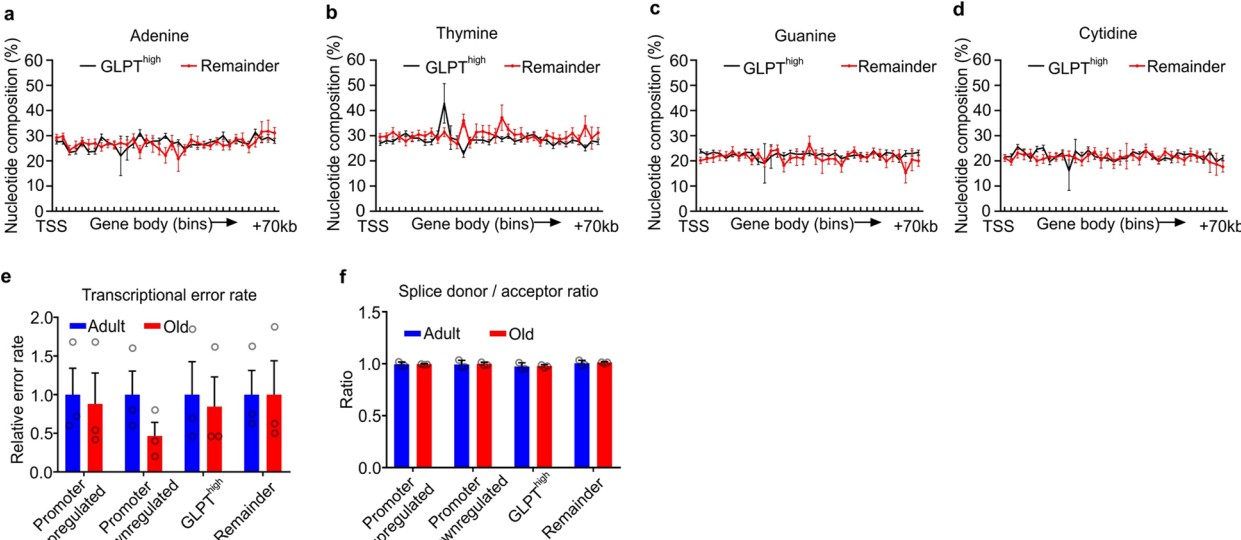

**Extended Data Fig. 4 | Correlation of transcriptional parameters to defined functional gene clusters, related to Fig. 4.** Transcriptional parameters in the identified functional gene clusters (as described in Extended Fig. 3a): **a–d**, average nucleotide composition per kb gene length in the template strand for the first 70 kb from TSS of top 50 GLPT[high] genes 70–80 kb (black line) compared to 50 remainder genes 70–80 kb that contain low GLPT levels (red line). Data are mean ± SEM. **e**, Transcriptional error rates in EU-seq data from wildtype adult (blue) and old (red) livers in 4 different gene sets: 'Promoter-upregulated' (n = 778), 'Promoter-downregulated' (n = 394), GLPT[high] (n = 914), remainder (n = 1884). Data are mean ± SEM. **f**, Bar diagram showing the ratio between EU-seq reads mapped to splice donor and acceptor sites of genes in each functional gene cluster in **e**. Average of n = 3 / group shown. Data are mean ± SD.

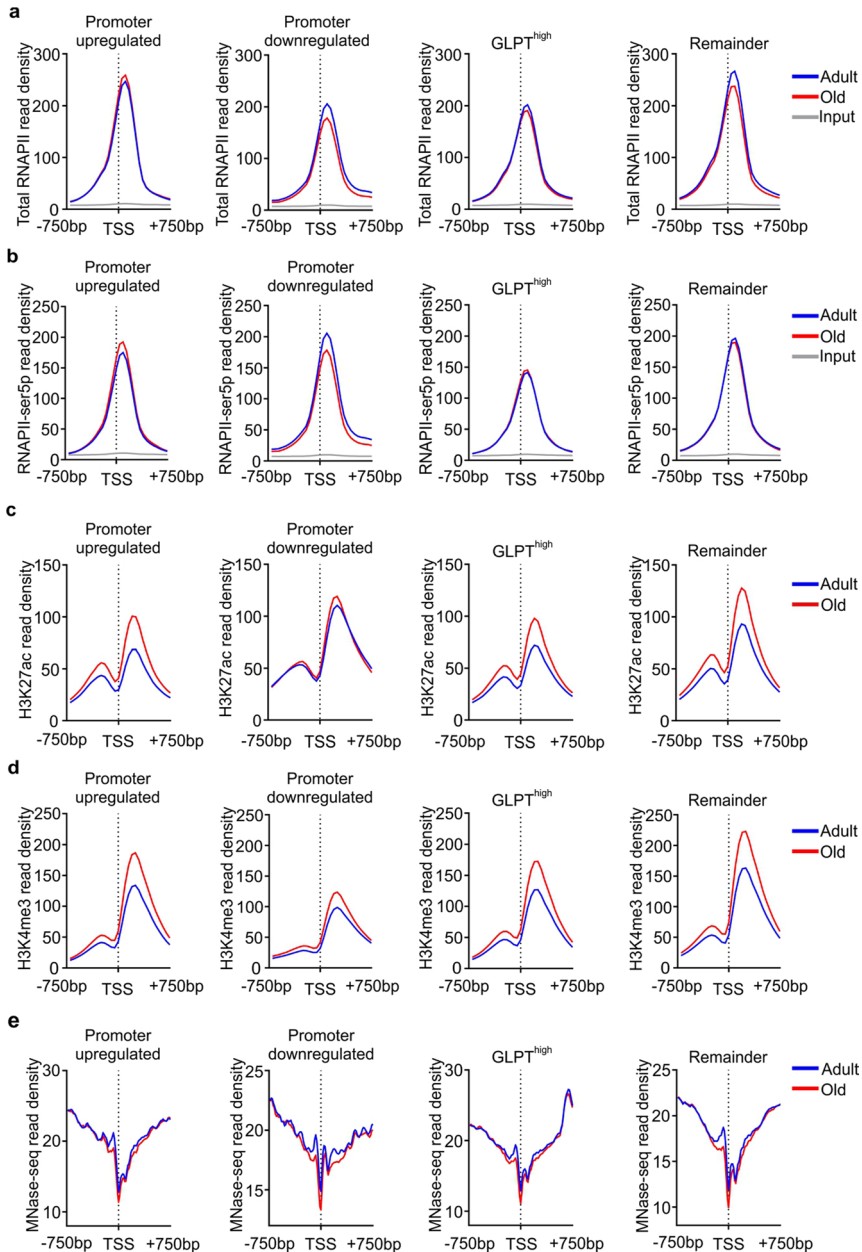

**Extended Data Fig. 5 | Levels of RNAPII or epigenetic markers around the TSS (±750 bp) of each defined functional gene cluster, related to Fig. 4.** Functional gene clusters: promoter-upregulated genes, promoter-downregulated genes, genes with a high gradual loss of productive transcription (GLPT^high) and remainder. Data are mean ± s.d. Blue lines represent adult liver, red lines represent old liver. Average of n = 3 / group shown for: **a**, Total RNAPII. **b**, serine 5 phosphorylated (ser5p) RNAPII. **c**, histone 3 lysine 27 acetylation (H3K27Ac; open chromatin). **d**, histone 3 lysine 4 trimethylation (H3K4Me3; open chromatin). **e**, inaccessible chromatin as MNase digests only DNA not bound to proteins including nucleosomes.

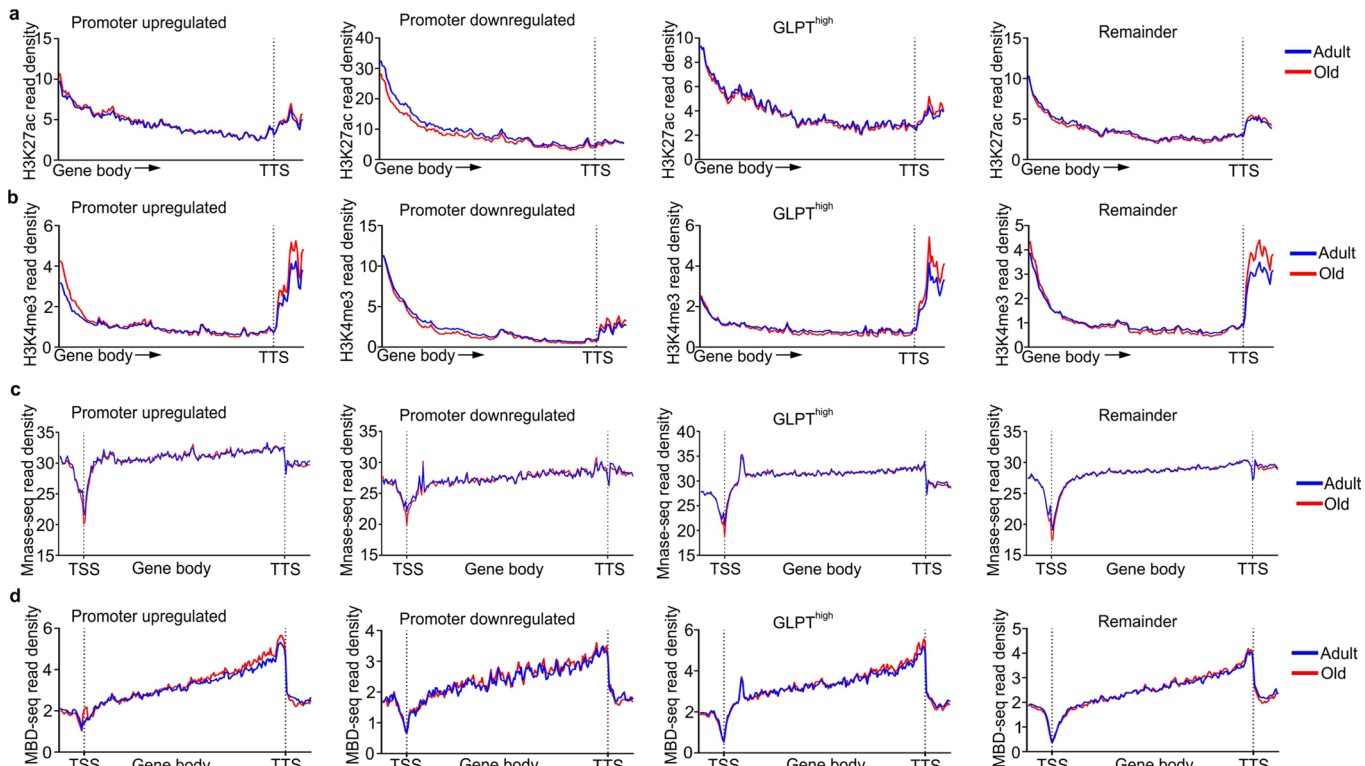

**Extended Data Fig. 6 | Levels of epigenetic markers throughout gene bodies of each defined functional gene cluster, related to Fig. 4.** Functional gene clusters: promoter-upregulated genes, promoter-downregulated genes, genes with a high gradual loss of productive transcription (GLPT[high]) and remainder. Blue lines represent adult liver, red lines represent old liver. Average of n = 3/ group shown of sequencing read density from TSS + 750 bp to TTS + 4 kb for: **a**, histone 3 lysine 27 acetylation (H3K27Ac; open chromatin). **b**, histone 3 lysine 4 trimethylation (H3K4Me3; open chromatin). **c**, inaccessible chromatin as MNase digests only DNA not bound to proteins including nucleosomes. **d**, DNA methylation status.

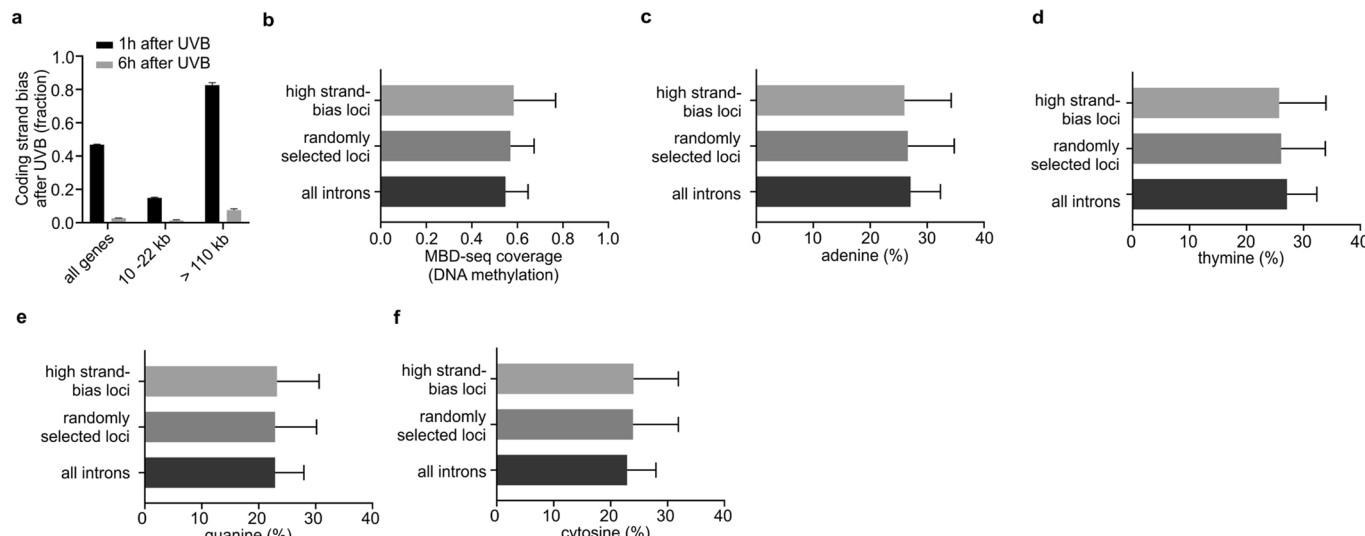

**Extended Data Fig. 7 | DNA damage-induced coding strand bias detection control, related to Fig. 4. a**, Bar diagram representing the coding strand bias in total RNAPII ChIP-seq data 1 hour and 6 hours after irradiating MCF7 cells with 55 J/m$^2$ UVB (data from[76]). All genes (n = 18224), short genes (10–22 kb) and long genes (>110 kb). Data are mean ± SEM. Note that the strand bias is only present in MCF7 cells 1 hour after UVB treatment, when RNAPII is still stalled on DNA lesions and DNA repair is ongoing. After 6 hours, most of the stalled RNAPII has been removed from the DNA lesions. This shows that i) the protocol used is able to detect a bias towards the coding strand and therefore can be used to analyze aging samples, ii) the coding strand bias is a transient phenotype after UVB. Based on published amounts of coding strand bias after a known UVC-

induced DNA lesion density[38], we estimate that livers from wildtype aged mice display a coding strand bias fraction in the range of 0.05–0.10. **b–f**, Mean local DNA methylation coverage (**b**) and (**c–f**) local nucleotide composition status in template strands of 50 genes with that exhibit the highest coding strand bias in general. The intragenic intronic region is chosen with the highest coding strand bias (high strand bias loci). This loci gene set is compared t i) random selected intragenic loci of similar size: 6 times 50 random intronic locations in the template strand, and ii) the complete intronic transcriptome; all introns from transcriptome (including high strand bias locations). Average of n = 50 / group shown. Data are mean ± SD.

# Reporting Summary

## Statistics

For all statistical analyses, confirm that the following items are present in the figure legend, table legend, main text, or Methods section.

| n/a | Confirmed | |
|---|---|---|
| ☐ | ☒ | The exact sample size (*n*) for each experimental group/condition, given as a discrete number and unit of measurement |
| ☐ | ☒ | A statement on whether measurements were taken from distinct samples or whether the same sample was measured repeatedly |
| ☐ | ☒ | The statistical test(s) used AND whether they are one- or two-sided<br>*Only common tests should be described solely by name; describe more complex techniques in the Methods section.* |
| ☒ | ☐ | A description of all covariates tested |
| ☐ | ☒ | A description of any assumptions or corrections, such as tests of normality and adjustment for multiple comparisons |
| ☐ | ☒ | A full description of the statistical parameters including central tendency (e.g. means) or other basic estimates (e.g. regression coefficient) AND variation (e.g. standard deviation) or associated estimates of uncertainty (e.g. confidence intervals) |
| ☐ | ☒ | For null hypothesis testing, the test statistic (e.g. *F*, *t*, *r*) with confidence intervals, effect sizes, degrees of freedom and *P* value noted<br>*Give P values as exact values whenever suitable.* |
| ☒ | ☐ | For Bayesian analysis, information on the choice of priors and Markov chain Monte Carlo settings |
| ☒ | ☐ | For hierarchical and complex designs, identification of the appropriate level for tests and full reporting of outcomes |
| ☒ | ☐ | Estimates of effect sizes (e.g. Cohen's *d*, Pearson's *r*), indicating how they were calculated |

*Our web collection on statistics for biologists contains articles on many of the points above.*

## Software and code

Policy information about availability of computer code

| Data collection | No software was used for data collection |
|---|---|
| Data analysis | Sequence-read quality control was performed by using Fastqc (v0.11.3; http://www.bioinformatics.babraham.ac.uk/projects/fastqc/)<br>Sequencing reads were pre-processed with Trimmomatic (v0.35) [PMID: 24695404]<br>RNA-seq reads were aligned by y using Tophat2 (v2.0.9) [PMID: 19289445].<br>Htseq (v0.6.0) [PMID: 25260700]  was used to quantify reads<br>Bowtie2 [PMID: 22388286] was used to align ChIP-seq reads.<br>FastQScreen (v0.14.0). https://www.bioinformatics.babraham.ac.uk/projects/fastq_screen/<br>FastQC (v0.11.9). https://www.bioinformatics.babraham.ac.uk/projects/fastqc/<br>K-means clustering software (Cluster 3.0; http://bonsai.hgc.jp/~mdehoon/software/cluster/software.htm)<br>HOMER v4.11 software for ChIP-seq profile generation [PMID: 20513432]<br>Nucleotide composition was determined by Quailmap (v2.21) [PMID: 26428292].<br>R  project for statistical computing. https://www.r-project.org/<br>EdgeR (DOI: 10.18129/B9.bioc.edgeR)<br>BioConductor seqTools (Kaisers W (2013). R package version 1.2.0.<br>IRanges packages (2.30.1) [PMID: 23950696].<br>Analysis of EU-seq read abundance at splicing donor and acceptor sites was carried out by a custom-written script:<br>Splicingdonor&acceptorfinder.py<br>Binning genes was carried out by a custom-updated script from the existing code from RSeQC (v3.0.1) geneBody_coverage.py:<br>K_binning.py<br>Ingenuity Pathway Analysis, QIAGEN.<br>Gene set enrichment analysis (GSEA). https://www.gsea-msigdb.org/gsea/index.jsp<br>Kyoto Encyclopedia of Genes and Genomes (KEGG). https://www.genome.jp/kegg/pathway.html |

Fiji (Image J 1.53q) https://fiji.sc/

Alternative splicing events were detected by Astalavista (v4.0) [PMID: 25577392]
Samtools (v1.9) [PMID: 19505943]
Bedtools (v2.27.1) [PMID: 20110278].
GC content and DNA methylation status were detected by Quailmap (v2.21) and deepTools (v2.0) [PMID: 27079975].
Graphpad Prism, Version 7.05. (WWW.graphpad.com)

For manuscripts utilizing custom algorithms or software that are central to the research but not yet described in published literature, software must be made available to editors and reviewers. We strongly encourage code deposition in a community repository (e.g. GitHub). See the Nature Portfolio guidelines for submitting code & software for further information.

## Data

Policy information about availability of data

 All manuscripts must include a data availability statement. This statement should provide the following information, where applicable:
- - Accession codes, unique identifiers, or web links for publicly available datasets
- - A description of any restrictions on data availability
- - For clinical datasets or third party data, please ensure that the statement adheres to our policy

The nascent RNA-seq and ChIP-seq data discussed in this publication have been deposited in NCBI's Gene Expression Omnibus, and are accessible through: https://www.ncbi.nlm.nih.gov/sra/PRJNA603447
The following publicly available data was used:
Mouse ribosomal DNA (BK000964.3) from NCBI.
Mouse mitochondrial sequences (UCSC, mm10) from UCSC genome browser.
Mouse reference genome (GRCm38/mm10) from Ensembl.
Human reference genome (hg19).
C.elegans reference genome (ce10).
Published datasets were downloaded from European Nucleotide Archive.
Total RNA-seq data from human tendon [PMID:25888722].
Total RNA-seq data from Caenorhabditis elegans [PMID: 29298683]
DNA methylation (PRJNA376757) [PMID: 28249716],
Histone H3K27ac and H3K4me3 data (PRJNA281127) [PMID: 30858345]
MNase-seq (GSE58005) [PMID: 25437555]
Reactome Pathway Databases, Version 79
Aging Perturbations from GEO down datasets from Enrichr
Kegg database, Release 99.0
QIAGEN Ingenuity Pathway Analysis

# Field-specific reporting

Please select the one below that is the best fit for your research. If you are not sure, read the appropriate sections before making your selection.

☒ Life sciences          ☐ Behavioural & social sciences          ☐ Ecological, evolutionary & environmental sciences

For a reference copy of the document with all sections, see nature.com/documents/nr-reporting-summary-flat.pdf

# Life sciences study design

All studies must disclose on these points even when the disclosure is negative.

| | |
|---|---|
| Sample size | Sample sizes were n=3 mice / group for all experiments. |
| Data exclusions | No data was excluded. |
| Replication | Experiments were performed at minimum of 3 times. All replicates were successful. |
| Randomization | Our study design did not require randomization. |
| Blinding | Our study design did not require blinding |

# Reporting for specific materials, systems and methods

We require information from authors about some types of materials, experimental systems and methods used in many studies. Here, indicate whether each material, system or method listed is relevant to your study. If you are not sure if a list item applies to your research, read the appropriate section before selecting a response.

## Materials & experimental systems

| n/a | Involved in the study |
|---|---|
| ☐ | ☒ Antibodies |
| ☐ | ☒ Eukaryotic cell lines |
| ☒ | ☐ Palaeontology and archaeology |
| ☐ | ☒ Animals and other organisms |
| ☒ | ☐ Human research participants |
| ☒ | ☐ Clinical data |
| ☒ | ☐ Dual use research of concern |

## Methods

| n/a | Involved in the study |
|---|---|
| ☐ | ☒ ChIP-seq |
| ☒ | ☐ Flow cytometry |
| ☒ | ☐ MRI-based neuroimaging |

# Antibodies

| Antibodies used | RNAPII-ser2p (Abcam, ab5095)<br>RNAPII-ser5p (Abcam, ab5131)<br>(Total) RNAPII RPB1-NTD (Cell signaling, D8L4Y)<br>Alexa-fluor-594-RPB1 (Biolegend, 664908)<br>Phospho-ATM (Ser1981, Cell Signaling #4526)<br>Phospho-Histone H2A.X (Ser139, Cell Signaling; #9718) |
|---|---|
| Validation | Validation for all antibodies is available at the manufacture's websites:<br>RNAPII RPB1-NTD (Cell signaling, D8L4Y) : https://www.cellsignal.de/products/primary-antibodies/rpb1-ntd-d8l4y-rabbit-mab/14958<br>RNAPII-ser2p (Abcam, ab5095) : https://www.abcam.com/rna-polymerase-ii-ctd-repeat-ysptsps-phospho-s2-antibody-ab5095.html<br>RNAPII-ser5p (Abcam, ab5131) : https://www.abcam.com/rna-polymerase-ii-ctd-repeat-ysptsps-phospho-s5-antibody-ab5131.html<br>Alexa-fluor-594-RPB1 (Biolegend, 664908) : https://www.biolegend.com/de-de/products/alexa-fluor-594-anti-rna-polymerase-ii-rpb1-antibody-12144<br>Phospho-ATM (Ser1981, Cell Signaling #4526) : https://www.cellsignal.de/products/primary-antibodies/phospho-atm-ser1981-10h11-e12-mouse-mab/4526<br>P-H2A.X (Ser139, Cell Signaling; #9718) : https://www.cellsignal.de/products/primary-antibodies/phospho-histone-h2a-x-ser139-20e3-rabbit-mab/9718 |

# Eukaryotic cell lines

Policy information about cell lines

| Cell line source(s) | mouse dermal fibroblasts isolated in the Erasmus MC experimental animal facility |
|---|---|
| Authentication | None of the cell lines used were authenticated |
| Mycoplasma contamination | Cell lines were confirmed mycoplasma free. |
| Commonly misidentified lines<br>(See ICLAC register) | *Name any commonly misidentified cell lines used in the study and provide a rationale for their use.* |

# Animals and other organisms

Policy information about studies involving animals; ARRIVE guidelines recommended for reporting animal research

| Laboratory animals | 15 and 104 weeks old male wild type mice in a F1 C57BL6J/FVB (1:1) hybrid background.<br>DNA repair deficient premature ageing mouse models in F1 C57BL6J/FVB (1:1) hybrid background:<br>4 and 10 weeks old male Ercc1Δ/− mutants [PMID: 9197240].<br>7 and 14 weeks old male Xpg-/- mutants [PMID: 25299392]. |
|---|---|
| Wild animals | No wild animals were used in the study. |
| Field-collected samples | No field collected samples were used in the study. |
| Ethics oversight | Animal experimentation followed the "Animal Welfare Act" of the Dutch government, named: the "Guide for the Care and Use of Laboratory Animals" as standard. License number: GGO 97-187, protocol 139-12-18, EMC number: 2767 |

Note that full information on the approval of the study protocol must also be provided in the manuscript.

# ChIP-seq

## Data deposition

☒ Confirm that both raw and final processed data have been deposited in a public database such as GEO.

☐ Confirm that you have deposited or provided access to graph files (e.g. BED files) for the called peaks.

| | |
|---|---|
| Data access links<br>*May remain private before publication.* | sra: PRJNA603447 |
| Files in database submission | RNA Polymerase ChIP-seq from liver (triplicates):<br>Total RNAPII ChIP-seq adult 1<br>Total RNAPII ChIP-seq adult 2<br>Total RNAPII ChIP-seq adult 3<br>Total RNAPII ChIP-seq old 1<br>Total RNAPII ChIP-seq old 2<br>Total RNAPII ChIP-seq old 3<br>RNAPII-ser2 ChIP-seq adult 1<br>RNAPII-ser2 ChIP-seq adult 2<br>RNAPII-ser2 ChIP-seq adult 3<br>RNAPII-ser2 ChIP-seq old 1<br>RNAPII-ser2 ChIP-seq old 2<br>RNAPII-ser2 ChIP-seq old 3<br>RNAPII-ser5 ChIP-seq adult 1<br>RNAPII-ser5 ChIP-seq adult 2<br>RNAPII-ser5 ChIP-seq adult 3<br>RNAPII-ser5 ChIP-seq old 1<br>RNAPII-ser5 ChIP-seq old 2<br>RNAPII-ser5 ChIP-seq old 3<br>Input ChIP-seq adult<br>Input ChIP-seq old<br><br><br>Nascent RNA-seq (EU-seq) from UV-irradiated cells:<br>EU-seq in vitro control (no UV)<br>EU-seq in vitro 2J/m2 UVC<br>EU-seq in vitro 4J/m2 UVC<br>EU-seq in vitro 6J/m2 UVC<br><br>Nascent RNA-seq (EU-seq) from progeria cohort:<br>EU-seq Ercc1Δ/- 4 weeks 1<br>EU-seq Ercc1Δ/- 4 weeks 2<br>EU-seq Ercc1Δ/- 4 weeks 3<br>EU-seq Ercc1Δ/-  10 weeks 1<br>EU-seq Ercc1Δ/-  10 weeks 2<br>EU-seq Ercc1Δ/-  10 weeks 3<br>EU-seq Xpg-/- 7 weeks 1<br>EU-seq Xpg-/- 7 weeks 2<br>EU-seq Xpg-/- 7 weeks 3<br>EU-seq Xpg-/- 14 weeks 1<br>EU-seq Xpg-/- 14 weeks 2<br>EU-seq Xpg-/- 14 weeks 3<br>EU-seq WT progeria cohort 4wk 1<br>EU-seq WT progeria cohort 4wk 2<br>EU-seq WT progeria cohort 4wk 3<br>EU-seq WT progeria cohort 14wk 1<br>EU-seq WT progeria cohort 14wk 2<br>EU-seq WT progeria cohort 14wk 3<br><br>Nascent RNA-seq (EU-seq) from normally aged cohort:<br>EU-seq WT adult liver 1<br>EU-seq WT adult liver 2<br>EU-seq WT adult liver 3<br>EU-seq WT old liver 1<br>EU-seq WT old liver 2<br>EU-seq WT old liver 3<br>EU-seq WT adult kidney 1<br>EU-seq WT adult kidney 2<br>EU-seq WT old kidney 1<br>EU-seq WT old kidney 2 |
| Genome browser session<br>(e.g. UCSC) | n.a. |

## Methodology

**Replicates**

Nascent RNA-seq and ChIP-seq was done in triplicates.

**Sequencing depth**

64-119M reads

**Antibodies**

RNAPII-ser2p (Abcam, ab5095)
RNAPII-ser5p (Abcam, ab5131)
RNAPII RPB1-NTD (Cell signaling, D8L4Y)

**Peak calling parameters**

Expressed genes were analyzed with annotated locations, thus no new peak calling was needed.

**Data quality**

Percentage of read position with a base quality score above 30 ranges between 90.32% - 93-31%

**Software**

Reads were mapped to Mus musculus genome version 10 (mm10) using Bowtie.
Read quality was assessed using FastQC and FastQScreen.
Annotated expressed genes were binned by K_bining.py.
Read coverage on annotated genes was quantified by HTseq.
Coverage profiles on genes was plotted by using HOMER software.

