## [Peer Review File · Nature Genetics]

Peer Review Information

Manuscript Title: Genome-wide RNA polymerase stalling shapes the transcriptome during aging

Corresponding author name(s): Joris Pothof

Reviewer Comments & Decisions:

Decision Letter, initial version:
--

10th Feb 2022

Dear Dr Pothof,

Your Article, "Genome-wide transcription stalling by DNA damage shapes the transcriptome in aging" has now been seen by 3 referees. You will see from their comments below that while they find your work of interest, some important points are raised. We are interested in the possibility of publishing your study in Nature Genetics, but would like to consider your response to these concerns in the form of a revised manuscript before we make a final decision on publication.

As you will see, Reviewer #1 calls this impressive work, but raises a series of technical concerns about the experiments and the controls, including the number of mice analyzed. Given that this is an aging study, we don't think that it is very reasonable to ask you to expand the cohort of mice. We are willing to overrule this point, but we recommend that you respond to the concern about number of mice, potentially pointing to recent papers in the area to justify your experimental design. The other questions and technical points should be addressed.

Reviewer #2 is overall supportive, but does raise a few points, including asking for orthogonal proof of DNA damage (also requested by Reviewer #1). We think that these requests are reasonable.

Reviewer #3 provides a very positive review with a comment for discussion.

In short, we think that there is a decent level of support here, and we will not require more mice as a condition of further consideration. However, the points about the controls and discussion of other interpretations of the data, as well as all other technical concerns, need to be thoroughly addressed.

We therefore invite you to revise your manuscript taking into account all reviewer and editor comments. Please highlight all changes in the manuscript text file. At this stage we will need you to upload a copy of the manuscript in MS Word .docx or similar editable format.

We are committed to providing a fair and constructive peer-review process. Do not hesitate to contact us if there are specific requests from the reviewers that you believe are technically impossible or

unlikely to yield a meaningful outcome.

*2) If you have not done so already please begin to revise your manuscript so that it conforms to our Article format instructions, available [here](http://www.nature.com/ng/authors/article_types/index.html). Refer also to any guidelines provided in this letter.

[redacted]

Also, please accept my apologies to you and your team that this round of review took so long. The delays were due to the time of year (holidays, grant deadlines) and we wouldn't expect similar delays

in subsequent rounds of review. I will do everything I can to ensure that the remainder of the process proceeds efficiently.

Thank you very much.

All the best,

Catherine

Catherine Potenski, PhD
Chief Editor
Nature Genetics
1 NY Plaza, 47th Fl.
New York, NY 10004
catherine.potenski@us.nature.com
<https://orcid.org/0000-0002-4843-7071>

Referee expertise:

Referee #1: transcription, aging

Referee #2: molecular mechanisms of aging, transcription

Referee #3: aging, gene regulation

Reviewers' Comments:

Reviewer #1:

Remarks to the Author:

In this manuscript the authors provide a possible explanation for aging-related changes in gene expression. They study this phenomenon in aged mouse liver using nascent RNA sequencing and RNA polymerase CHIP-seq. Some of their most striking conclusions include: decreased total transcriptional output in aged liver, a decline in nascent transcription and increase in polymerase density towards the end of the gene body, the ability to replicate this phenomenon with DNA repair deficient mice, evidence of polymerase-stalling DNA lesions from a coding strand bias in CHIP-seq, and by computational analysis their measure of transcription stress being a strong predictor of age-related changes in gene expression. From these findings they claim that DNA damage is responsible for a majority of changes in the aged transcriptome. The work is impressive and some of the conclusions are convincing, however some decisions in experimental design may be causing artifacts that need to be addressed. The following comments outline suggested improvements to the manuscript:

General issues

1. Mice sample size (n=3) is small. It is hard to interpret the significance of the results unless there is

closer to 10 mice per group. The experiments with another mice strain where both males and females are included will add confidence to the generality of the results.

2. Chip-seq results are impressive and indeed demonstrate the reduced mRNA synthesis in older animals. However, the current experiments do not actually explain why RNAP is found more often closer to the end of a gene.

The authors suggestion that RNAPII stalls at DNA damage sites in old mice is quite speculative. There could be many reasons for RNAPII to spend more time in gene body and incorporate less 5-EU.

The experiments suggested below should help to validate the authors' conclusions:

Specific points:

1. The authors need to actually demonstrate the increased DNA damage in the experimental old mice. It is not sufficient to make a reference to some previous papers or to demonstrate it in a genetic model that is prone to much higher DNA damage. In xpg animals the mechanism could be different from natural aging. The claims of accumulated DNA damage in aged liver rely on indirect evidence by a coding strand bias in CHIP-seq. Some other measure of DNA damage would be appropriate.
2. A possible explanation for the authors' observations would be that the rate of transcription elongation just declines with age (regardless of DNA damage). That could be a result of decreased RNAP activity or substrate (NTPs) pools in old animals. The authors need to address these possibilities.
3. I am puzzled with the decreased 5-EU incorporation only in the introns of RNAPII transcribed genes. Does it mean there is less/no DNA damage in exons and/or genes transcribed by other polymerases? Without RNA fragmentation the 5-EU incorporation will be biased towards the 5' ends and decrease the randomness of random primer annealing (PMID: 18516045)–. That might explain the decrease of reads density along the gene length. The authors should address this possible artifact.
4. To claim that there is less transcriptional output in aged liver, one needs to show that differential EU uptake is not a factor in the observed EU signal in old liver. How is it that there is "identical EU incorporation densities in adult and old livers" in extended data Fig 2D, but "1.5-fold reduced EU signal in old compared to adult liver" in Fig 1A? It is not satisfactory to just see the amount of cDNA produced being the same, as this readout has been affected by several manipulations.
5. A similar argument is made in Fig 1G. The authors find that EU incorporation is only affected in reads originating from RNAPII and use this as evidence for similar EU uptake or metabolism. In order to make this argument, the authors would need to claim that DNA damage is only occurring on DNA accessible to RNAPII and not RNAPI, RNAPIII, or RNAP-MT. This is unlikely and no evidence is provided. Further explanation is needed for why EU incorporation is only changed with RNAPII.
6. EU-seq appears to be very similar to 4SU-seq and likely suffers from the same bias. Namely, it is not truly representative of nascent RNA since it is biased toward the 5' end of RNA. This phenomenon is addressed in TT-seq (PMID: 27257258) with a fragmentation step prior to the pulldown of labeled RNA. Because of this, it is difficult to determine whether the decline in transcription at the 3' end of a gene body is due to transcriptional stress and/or just an artifact of EU-seq. This artifact would only be magnified in longer genes.

7. It does appear that the decline in nascent RNA at the end of gene bodies is magnified in aged liver, however, because of the prior concerns with EU uptake and potential EU-seq artifacts, it is not very unconvincing. One alternative explanation is that the slower EU uptake shortens the labelling time in aged liver. This could bias EU-seq towards the observed phenotype.
8. If the authors are claiming that accumulated DNA lesions cause a majority of problems with elongation, it would be interesting to see a rescue experiment showing how nascent RNA across gene bodies changes when fewer lesions are expected. Perhaps in younger mice or with some manipulations to the MDF cells. It appears there is data from wild-type mice at 7 weeks. Is there any difference in 'GLPT' between wild-type mice at 7 and 15 weeks?
9. Under the conditions of persistent stalling, RNAP II is known to be rapidly degraded via ubiquitination. The authors need to explain the persistence of stalled RNAP II molecules in aged mice.
10. The authors demonstrated a dose dependent transcriptional decline in quiescent ERCC1 d/- cells after 24 hrs recovery. It would be nice to have side by side comparison on wt cells at the same doses and same recovery time. Also another time point with longer recovery time is needed to evaluate whether these damage and transcription blocks are fixed in the cells.

Reviewer #2:

Remarks to the Author:

In the manuscript titled "Genome-wide transcription stalling by DNA damage shapes the transcriptome in aging", the authors, Gyenis et al., report that age-associated transcriptional stress caused by DNA damage results in transcriptional stalling, altering the transcriptome in aged cells. They used novel sensitive nascent RNA sequencing to detect abnormalities in transcription in aged mouse liver and correlated them with RNA polymerase II stalling mapped by ChIP-seq. Finally, similar stalling and transcriptional defects were also detected in mouse models with defects in DNA repair, suggesting accumulated DNA damage during aging is likely the underlying cause.

Overall, this project is carefully designed and executed, most results are convincing and support the major conclusions. I find that a few more clarifications will make this manuscript acceptable for publication.

1. Consideration for EU labeling period and controls for optimization and labeling efficiency/equality between young and old animals.
2. Given the similarity in RNA polymerase II stalling between aged mice and mice with DNA repair defects (Xpg^{-/-} and Ercc1^{-/-}), it is reasonable to postulate that DNA damages are also causing RNA polymerase II stalling in aged mice. However, since DNA damages causing age-related RNAP II stalling is a key point in this manuscript, it should be further corroborated. For example, can DNA damage signals (like gamma-H2A.X) can be detected in RNAP II stalling sites in aged mice? Are DNA damage and RNAP II stalling reduced in any of the long-lived mouse models (either dwarf mouse models or known longevity interventions)?
3. Figure 4h showed coding strand bias in aged mice? How do they compare to young mice?
4. In Figure 5a-b, the authors showed nascent RNA-seq profiles for Igf1 gene, making the point that transcriptional stress affects age-related pathways. It is also important to corroborate this point with RNAP II ChIP-seq data at Igf1 gene to directly show that RNAP II stalling is occurring in aged mice.

Reviewer #3:

Remarks to the Author:

During the recent years, the research field on ageing has witnessed much and compelling evidence – mainly from the Hoeijmakers group- supporting the notion that persistent DNA damage drives aging. Further work revealed that the gradual accumulation of irreparable DNA lesions in the mammalian genome impinges selectively on pathways or processes that causally contribute to age-related diseases. However, the question remained how or why gene expression programs change when DNA damage accumulates over long periods of time, which genes or pathways are at greater risk or why certain types of cells or tissues are more vulnerable to persistent DNA lesions than others.

The present work is paradigm-shifting; it provides compelling, in vivo evidence that DNA damage interferes with the process of ongoing mRNA synthesis. The authors provide multiple evidence that transcription initiation remains unaltered with no difference on RNAPII recruitment on TSS or the transition of RNAPII from promoters to the main gene body. Strikingly, unlike the higher phosphorylated RNAPII levels and the functionally intact promoters, the authors find that nascent RNA synthesis gradually declines in highly transcribed genes with aging. The decline in gene transcription is dependent on primary gene length (and time). Using rapidly aging animals or mouse dermal fibroblasts with engineered defects in NER, the authors show that the severity of the DNA repair defect itself or the levels of oxidative or UV-induced DNA damage correlate well with the reduced transcription elongation. Importantly, the authors show that in old livers, there is coding strand bias, which also depends on gene length; that is, the DNA lesion in the template strand that stalls RNAPII also impairs DNA amplification of that strand. The authors then move on to provide evidence that their findings are universal and relevant to multiple species and organs with dramatically different developmental or functional requirements.

Overall, this compelling work should be published as is with no further experiments. The experiments are carefully designed and the conclusions are well supported from the primary observations. To further improve the content, the authors could reflect on whether exposure of mice to calorie restriction is expected to minimize the RNAP2 stalling frequencies in aged mouse livers.

Author Rebuttal to Initial comments

Dear editor,

We would like to thank the reviewers for their positive reviews, their uniform agreement that our work will add substantial new information to the field of aging biology and their constructive criticism to improve our manuscript. We have addressed all points raised by each reviewer hereunder and look forward to your response.

Best regards,

Joris Pothof

Reviewer #1:**Remarks to the Author:**

In this manuscript the authors provide a possible explanation for aging-related changes in gene expression. They study this phenomenon in aged mouse liver using nascent RNA sequencing and RNA polymerase CHIP-seq. Some of their most striking conclusions include: decreased total transcriptional output in aged liver, a decline in nascent transcription and increase in polymerase density towards the end of the gene body, the ability to replicate this phenomenon with DNA repair deficient mice, evidence of polymerase-stalling DNA lesions from a coding strand bias in CHIP-seq, and by computational analysis their measure of transcription stress being a strong predictor of age-related changes in gene expression. From these findings they claim that DNA damage is responsible for a majority of changes in the aged transcriptome. The work is impressive and some of the conclusions are convincing, however some decisions in experimental design may be causing artifacts that need to be addressed. The following comments outline suggested improvements to the manuscript:

General issues

1. Mice sample size (n=3) is small. It is hard to interpret the significance of the results unless there is closer to 10 mice per group. The experiments with another mice strain where both males and females are included will add confidence to the generality of the results.

Indeed, the reviewer is correct that more animals and use of another mouse strain will increase quantitative significance of the results. However, the significance of the findings reported here can be independently and even more convincingly deduced from the fact that transcription stress is found to be a universal phenomenon, conserved over large evolutionary distances from worms to mammals (Figure 6A), occurs in many organs and tissues and explains the majority of aging-related transcription changes in public database of numerous species and organs/tissues (Figure 6C). This shows that

e.g. adding another mouse strain with a different genetic background would not much added value, as the phenomenon of transcription stress is even observed in worms.

Secondly, we agree that larger numbers of replicates will increase the power of the study, but this also depends on the degree of intrinsic variability in the experimental set-up. Larger numbers are especially relevant for differentially expressed gene identification (mRNA-sequencing) in which the number of data points per gene is the same as the number of replicates in an experimental group. In case of n=3, each gene

will have three data points and larger experimental groups will better identify significantly regulated genes. However, low number of replicates for mRNA expression analysis experiments are quite commonly used, also in high impact journals. For example, Schaum et al., Nature, 2020 performed a large-scale mRNA sequencing analysis to map gene expression patterns in multiple organs in aging (n=2-4), MaesoDíaz et al., Aging Cell, 2022 (n=3 / condition; mRNA-Seq), Micheli et al., Front Cell Dev Biol, 2021 (n=2 / condition; mRNA-Seq), Jin et al., Nat Neurosci, 2021 (n=3 / condition; total RNA Seq).

In our nascent RNA sequencing and ChIP-Sequencing experiments, we do not select for significant genes, but for intragenic transcriptional patterns across all expressed genes. Hence, each data point consists of the average of thousands of genes in three mice per group, regardless of their direction of regulation or significance. This type of analysis for intragenic transcriptional patterns are commonly published with lower numbers of replicates than n=3. Examples are Saponaro et al., Cell, 2014 (n=1 per condition ChIP-Seq; nascent RNA Seq), Geijer et al., Nat. Cell. Biol., 2022 (n=2 per condition; nascent RNA seq), Cugusi et al., Mol. Cell, 2022 (n=2 per condition; nascent RNA seq), Nakazawa et al., Cell, 2020 (n=1-3 per condition; RNAPII ChIP-Seq).

As we were not able to produce additional EU-labelled aged mouse livers, we also performed nascent RNA sequencing (EU-Seq) on young and old (2 years) mouse kidney, in which, as expected, we also observed transcriptional decline in aging. We have added this data to Figure 6a. Taken all data together, there is no doubt about the reproducibility and about the significance of the performed experiments.

2. Chip-seq results are impressive and indeed demonstrate the reduced mRNA synthesis in older animals. However, the current experiments do not actually explain why RNAP is found more often closer to the end of a gene.

Indeed, we did not explain this. A study using a genome-wide sequencing approach to map DNA excision repair shows that transcription-coupled repair, which is needed to remove stalled RNAPII from the lesion is exclusively active in transcribed strands and is most effective at the beginning of genes, where ongoing transcription is also most intense (Hu et al., Genes Dev, 2015; PMID: 25934506). Next to providing this explanation in the text, we also added an additional analysis demonstrating that the coding strand bias in aging, which measures polymerase-blocking lesions in template strands in isolated, pure DNA from RNAPII ChIP-Seq experiments, is more pronounced in 3' ends of genes, thereby further providing evidence for this scenario.

The authors suggestion that RNAPII stalls at DNA damage sites in old mice is quite speculative. There could be many reasons for RNAPII to spend more time in gene body and incorporate less 5-EU.

We agree with this statement, which is the reason that we performed the coding strand bias in aging analysis (Figure 4h-j, as developed in: <https://doi.org/10.1016/j.cell.2020.02.010>, cited in the manuscript), which tests the hypothesis that in isolated, pure genomic DNA from a RNAPII CHIP an increase in the number of perturbations occurs that block polymerase activity in the template strand. Since the amplification is done with purified DNA, the finding that the template strand is less well amplified compared to the coding strand must be due to the fact that the purified template strand DNA is damaged: conversely, intact DNA would have allowed equal amplification. In figure 5A depicting read distribution over the *Igf1* gene, we now also added RNAPII level distribution and the coding strand bias in aging. The first onethird of the gene body does not exhibit significant transcription loss and RNAPII increase, but the last two-thirds do. Here we clearly see a correlation between coding strand bias in aging and transcriptional decline, indicating that DNA damage is the culprit. The striking similarities with DNA damage induced by UV, endogenous DNA damage accumulation in long-term cell culture, the studies in the different repair mutants as well as the testing of various alternative hypotheses (Extended data Figure 4-6) is why we eliminated many of the other possible scenarios. Other explanations as raised by the reviewers are addressed below.

The experiments suggested below should help to validate the authors' conclusions:

Specific points:

1. The authors need to actually demonstrate the increased DNA damage in the experimental old mice. It is not sufficient to make a reference to some previous papers or to demonstrate it in a genetic model that is prone to much higher DNA damage. In xpg animals the mechanism could be different from natural aging. The claims of accumulated DNA damage in aged liver rely on indirect evidence by a coding strand bias in CHIP-seq. Some other measure of DNA damage would be appropriate.

This point raised is very understandable (and also raised by reviewer #2), but also technically much harder than generally thought. Unfortunately, outside of the DNA repair field there is a wide misunderstanding that accurately detecting physiological levels of endogenous DNA damage, that is highly heterogeneous in nature and each type of lesion present in minute quantities can be done. This is the reason we incorporated a separate paragraph in our recent Nature Review (Schumacher et al., Nature 2021; PMID: 33911272) devoted to this topic which lists all the shortcomings with current technologies for quantitative detection of physiological DNA damage. While for few endogenous DNA

lesions reliable assays have been developed, such as for double strand breaks by γ H2AX staining or for the oxidative damage 8-oxo-dG. While DSBs are present in very low quantities (<3 endogenous breaks / cell), which cannot explain the transcription stress phenotype, 8-oxo-dG lesions do not lead to transcription stalling. Hence, it is important to measure the correct type of damage. Moreover, in all likelihood there is not one type of DNA injury that is responsible for transcription stalling, but very probably a wide variety of very different types of lesions, including DNA-DNA and DNA-protein crosslinks and numerous types of distorting chemical modifications and adducts that interfere with proper base-pairing or physically block the polymerase. Finally, there is no method available that is able to determine the DNA lesion that actually blocks a RNA polymerase. The references cited in the manuscript quantify endogenous transcription-blocking lesions (e.g. cyclopurines, aldehyde damages, advanced glycation endproducts) that occur genome-wide in normal aging and demonstrate elevated levels in aging up to a point that they can explain our results.

One of the aims when we started the project leading to this manuscript was not only providing additional evidence for transcription-blocking lesions in normal aging, but also their functional consequences, i.e. that endogenous transcription-blocking DNA damage results in transcription stress. As DNA damage under normal physiological conditions is present in low quantities and heterogeneous in nature, there was a need to include an assay that detects all of them simultaneously, preferably in the gene body's template strand. Actually, this is why we are very happy with our assay, which measures the coding strand bias in aging, which is able to detect a wide variety of DNA lesions that interrupt polymerase activity and thus also transcription in the template strand of the transcribed compartment of the genome. The coding strand bias measurements are in itself already an assay detecting transcription-blocking DNA damage genome-wide.

In addition, we searched for DNA damage markers that would further provide evidence for transcription blocks by DNA damage and subsequent damage signaling. To our knowledge, the only known DNA damage signal that marks stalled transcription by DNA damage is non-canonical phosphorylation of DNA damage checkpoint kinase ATM, in which non-canonical refers to ATM activation in the absence of double strand DNA breaks (Tresini et al., Nature 2015; PMID: 26106861). We tested non-canonical phosphorylation of ATM in young and old liver and found a significant increase in aging, which further substantiates our scenario of DNA damage-induced transcription stress in aging. We have added this data to Figure 4 and thank the referee for this comment.

2. A possible explanation for the authors' observations would be that the rate of transcription elongation just declines with age (regardless of DNA damage). That could

be a result of decreased RNAP activity or substrate (NTPs) pools in old animals. The authors need to address these possibilities.

A BioRxiv manuscript indicates a ~5% increase in transcription speed in naturally aged mouse liver (Debes, et al., BioRxiv, 2019; <https://doi.org/10.1101/719864>). In the reasoning of the reviewer this would lead to an increase rather than a decline of transcriptional output in aging and hence, a change in transcription speed is likely not responsible for the observed transcription stress phenotype, which involves a 40% decrease. Moreover, altered transcription speed with aging would not explain the prominent gene-length bias, that we (and others) demonstrate. Rather accumulation of stochastic DNA damage perfectly explains this phenomenon.

Concerning the possibility of a shortage of a single NTP with aging this could also cause a transcriptional delay at gene regions that have a high content of that specific NTP. To that end, we had added in Extended data figure 4a-d the nucleotide content across gene bodies that does not differ between genes with high transcription stress levels, up- or down-regulated genes by promoter activity or the remainder. To provide additional evidence, we now also added local NTP content in regions with a high coding strand bias in aging (Extended data figure 7c-f). Also in this scenario, we did not find any overrepresentation, indicating that NTP pool levels are likely not involved in the transcription stress phenotype. Additionally, as mentioned in manuscript, the NTP-pool effect is counter-argued by our finding that RNAPI, RNAPIII and mt-RNAP transcription are not affected (Extended data figure 2h) and that the EU densities in nascent RNA are the same for young and old (Extended data figure 2d-g).

3. I am puzzled with the decreased 5-EU incorporation only in the introns of RNAPII transcribed genes. Does it mean there is less/no DNA damage in exons and/or genes transcribed by other polymerases?

We would like to address this misunderstanding. Figure 5A-C clearly shows that transcription also declines across exons. However, it is not present when displaying all reads mapping to exons (Figure 1h). We were first also puzzled by this observation, but as figure 5a,b demonstrates is that sequence reads mapping to exons in the beginning of genes could also be increased, while a loss is observed in exons at the end of the gene body. This could compensate when we only plot total reads mapping to exons and not their intragenic pattern.

Target RNA species of RNAPI, RNAPIII and RNAP-MT are relatively, resp. very small in comparison to the average size of RNAPII genes. For example, ribosomal RNA species are 121, 156, 1869 and 5070 nucleotides synthesized by RNAPI as a 47S precursor of 13.314 bp. Small nucleolar RNA species are around 200 nucleotides, tRNAs around 70 nucleotides. The mitochondrial genome is 16.5 kb in total, encoding 13 protein coding genes, 2 ribosomal RNAs and 22 tRNAs. With an estimated decline of 0.35%

of transcription every kb DNA and 1.5 DNA lesions every 100 kb, the putative decline in RNAPI, RNAPIII and RNAP-MT target genes is too small to detect with the used techniques. We have added this explanation in the manuscript for clarification (page 9, lines 222-224).

Without RNA fragmentation the 5-EU incorporation will be biased towards the 5' ends and decrease the randomness of random primer annealing (PMID: 18516045). That might explain the decrease of reads density along the gene length. The authors should address this possible artifact.

Decrease of read density along the gene length is indeed observed with EU-Seq in both adult and old mice as well as cell culture. This is not an artifact. This is not only because of the absence of RNA fragmentation, but also the EU incubation time in combination with the directionality of transcription. We have addressed this point in more detail at remark #6.

4. To claim that there is less transcriptional output in aged liver, one needs to show that differential EU uptake is not a factor in the observed EU signal in old liver. How is it that there is “identical EU incorporation densities in adult and old livers” in extended data Fig 2D, but “1.5-fold reduced EU signal in old compared to adult liver” in Fig 1A? It is not satisfactory to just see the amount of cDNA produced being the same, as this readout has been affected by several manipulations.

We already observed and mentioned that cDNA fragment length and length distribution (but not quantity as stated by the reviewer) is almost identical in our protocol. Since reverse transcriptase cannot synthesize over an EU covalently bound to a biotin molecule, cDNA fragment size in itself already suggests similar EU incorporation frequencies (Extended data figure 2D). The only manipulation was the isolation of total RNA followed by nascent RNA isolation and cDNA synthesis.

In addition, we added more proof about labeling efficiency / equality between young and old animals. If there is a ~1.5-fold reduced availability of EU in the cellular rNTP pool, then there would be ~1.5-fold less incorporation of EU into nascent RNAs, i.e. the distance between incorporated EU is likely also 1.5-fold increased. As our EUSeq protocol isolates all nascent RNAs with at least one EU incorporated, a 1.5-fold reduction in EU incorporation should especially be visible in very small RNA species (<300 nucleotides) in which we would expect a decline in the percentage of reads mapping to those small RNA species. However, we measured a non-significant and modest increase in the percentage mapped reads in small RNA species in aging, indicating equal EU availability (Extended data figure 2). As a control, we also analyzed the percentage reads mapping to RNA species with a length of 1-3 kb or 2-4 kb. Also in these categories, we do not observe differences between young and old mice.

Next, we also calculated the probabilities that at least one EU is incorporated into a nascent RNA molecule for different EU incorporation rates and therefore available for sequencing. Note that EU-biotinylation is likely saturated as there is about >10x biotin available for every EU nucleotide in the reaction. The probability that 1.5-fold less available EU and concomitant less incorporation into small nascent RNA species results in similar percentages of mapped reads to those small RNA species is $p < 2.2 \times 10^{-16}$

(Mann-Whitney U test, where p is the smallest p value that R scripts can provide). As a control, we also calculated the probability of EU incorporation in RNA species of 1-3 kb and 2-4 kb length, which was consistently large ($p > 0.99$ in both cases), indicating we would expect equal representation of these classes in the sequence data from young and old mice. Full detail of mathematical and statistical reasoning behind the calculations above is available in the extended data (Extended data figure 2e-g).

In conclusion, the scenario in which reduced EU availability leads to the observed results is highly unlikely.

5. A similar argument is made in Fig 1G. The authors find that EU incorporation is only affected in reads originating from RNAPII and use this as evidence for similar EU uptake or metabolism. In order to make this argument, the authors would need to claim that DNA damage is only occurring on DNA accessible to RNAPII and not RNAPI, RNAPIII, or RNAP-MT. This is unlikely and no evidence is provided. Further explanation is needed for why EU incorporation is only changed with RNAPII.

We did not clarify this in our manuscript. Target RNA species of RNAPI, RNAPIII and RNAP-MT are very small in comparison to the average gene size of structural genes transcribed by RNAPII (as mentioned above for point 3). We have added this explanation in the manuscript for clarification and are grateful to the referee for raising this point.

6. EU-seq appears to be very similar to 4SU-seq and likely suffers from the same bias. Namely, it is not truly representative of nascent RNA since it is biased toward the 5' end of RNA. This phenomenon is addressed in TT-seq (PMID: 27257258) with a fragmentation step prior to the pulldown of labeled RNA. Because of this, it is difficult to determine whether the decline in transcription at the 3' end of a gene body is due to transcriptional stress and/or just an artifact of EU-seq. This artifact would only be magnified in longer genes.

We agree that EU-Seq, like any method for labeling and isolating intact nascent RNA, exhibits an intrinsic bias towards the 5' beginning, as this part is present in any growing RNA molecule, whereas the 3'-end is

only present when the RNA polymerase has reached the TTS. We would like to stress that we did not carry out a fragmentation step and hence this is not an artifact, but intrinsic to an intact nascent RNA molecule, which is the complete pre-mRNA that is being produced in a specific time frame and not only the transcriptional front that one sequences if fragmentation is performed prior to nascent RNA isolation (in combination with very short nascent RNA labelling time). Since transcription is directional from the 5' beginning, there is always an "RNA tail" present and the longer the gene, the more decline one sees towards the 3' end. This is also observed in young mice or cell cultures and has been explained in the manuscript (Fig. 3b, page 6, lines 151-154). However, transcription in the beginning of genes is similar between adult and old mice, indicating that the samples from both will have the same technical biases. In all our analyses, we always first compare per gene between adult and old mice (e.g. Figure 3), and therefore, gene-specific, context and technical biases are reduced as much as possible.

Please note that the indicated TT-Seq method not only displays such a bias because of fragmentation, but also because of a very short (5 minute) nascent RNA labelling time (which is not feasible in mice) that only labels the very transcriptional front and not the tail. It can already be predicted that when labelling time increases in TTSeq, also the bias towards the 5' beginning of genes will emerge as transcription is directional and nascent RNA molecules will be labelled also upstream of the transcriptional front.

7. It does appear that the decline in nascent RNA at the end of gene bodies is magnified in aged liver, however, because of the prior concerns with EU uptake and potential EU-seq artifacts, it is not very unconvincing. One alternative explanation is that the slower EU uptake shortens the labelling time in aged liver. This could bias EU-seq towards the observed phenotype.

Slower uptake leads reduces EU incorporation time. This will not bias the EU-Seq phenotype with regard to its declining slope as elongating RNAPII is stochastically distributed over gene bodies, both across genes and cells. It does alter however, the sequencing read ratio between adjacent exons and introns as more labeled mRNAs are also sequenced, which do not contain intronic regions. In that sense, total RNA sequencing, which sequences all RNA species without selection can be regarded as nascent RNA sequencing with very long labelling times.

As seen in the comparison of total RNA and nascent RNA sequencing in extended data figure 2C, the sequencing read ratio between adjacent exons and introns in total RNA seq is as expected larger than in nascent RNA sequencing. The slope is not altered.

Secondly, slower EU uptake will lead to less labelled nascent RNA in the same time frame, which will be especially visible in the small RNA species category as that would also lead to less sequencing

reads mapped. As we do not see any differences (see also explanation at point #4), we have no reason to think that a slower uptake is responsible.

Finally, we would like to note that we also observe the transcription stress phenomenon by analyzing total RNA sequencing data sets that contain a significant fraction of reads mapping to intronic RNA (see Fig 6a), which does not involve any nascent RNA labeling and selection.

8. If the authors are claiming that accumulated DNA lesions cause a majority of problems with elongation, it would be interesting to see a rescue experiment showing how nascent RNA across gene bodies changes when fewer lesions are expected. Perhaps in younger mice or with some manipulations to the MDF cells.

Rescue experiments are conceptually difficult to establish. Improving DNA repair is notoriously difficult (also discussed in our Nature review (Schumacher et al., Nature,

2021; PMID: 33911272)) as this is a complex multi-protein and multi-pathway process.

Overexpression of a single factor does not lead to improved DNA repair and is sometimes even toxic. mRNA expression loss of the very long genes for Igf1 receptor (281 kb) and growth hormone receptor (265 kb) after UV-treatment has been published (Garinis et al., Nature Cell Biology, 2009; PMID: 19363488). This publication also included a rescue experiment in which cells were ectopically supplemented with UV photolyases. Photolyases are a one-protein DNA repair process that repairs a UV-induced DNA dimer using the energy of blue light. Photolyase genes are lost in evolution in placental mammals. This rescue experiment indeed restored the mRNA expression of the *Igf1* receptor and growth hormone receptor. In addition, we published that calorie restriction, one of the most robust longevity-promoting interventions and also dramatically improves health and lifespan in DNA repair mutant premature aging mouse models, restores the expression of mRNAs originating from long genes (Vermeij et al., Nature, 2016; PMID: 27556946), indicating that mice that are biologically younger do not exhibit a loss of long gene expression phenotype. We have added this information to the discussion.

It appears there is data from wild-type mice at 7 weeks. Is there any difference in 'GLPT' between wild-type mice at 7 and 15 weeks?

There are no significant and notable differences in 7- and 15-week-old wildtype mice with regard to GLPT.

9. Under the conditions of persistent stalling, RNAP II is known to be rapidly degraded via ubiquitination. The authors need to explain the persistence of stalled RNAP II molecules in aged mice.

Currently, we do not know the fate and molecular details of the stalled RNAPII complexes in aging. This is subject of ongoing research as we cannot exclude slow turnover and a semi-dynamic situation in which the RNAPII stay arrested for significant time but at some point, are either helped by (still functioning) TCR or are degraded. But the finding that in old hepatocytes 40% of elongating RNAPII is stalled, is certainly not favoring the idea of substantial RNAPII degradation, which then should be compensated by a high turnover. Most data indicating RNAPII ubiquitination and degradation have been obtained in cultured cells, which are proliferating, involving replication/transcription conflicts that might trigger stalled RNAPII complexes to be degraded. This is the reason why in the *in vitro* culture experiments (Fig 4e-g) we used quiescent cells. Moreover, numerous processes are altered in aged mice, which could influence RNAPII ubiquitination and degradation parameters, which complicates speculation. For example, several studies, including this manuscript, shows that ubiquitin-mediated proteolysis and proteasome-mediated degradation are down-regulated or rewired (Koyuncu et al., Nature, 2021; PMID: 34321666; Sacramento et al., Mol. Syst. Biol. 2020; PMID: 32558274), which might explain the persistence of stalled RNAPII.

10. The authors demonstrated a dose dependent transcriptional decline in quiescent ERCC1 d/- cells after 24 hrs recovery. It would be nice to have side by side comparison on wt cells at the same doses and same recovery time. Also another time point with longer recovery time is needed to evaluate whether these damage and transcription blocks are fixed in the cells.

We did not include longer time points. One hallmark of cells with a defect in transcription-coupled repair (TCR) is that there will never be a recovery, also not after 48 hours. This property is used since the late 1980's as a clinical diagnostic test for rare TCR syndromes. This is also the reason we did not include wildtype cells in this analysis. Moreover, we chose the 24h time point as at that time TCR mutant cells are still healthy, but more importantly transcription properties are restored. It is known that immediately after UV damage, which delivers the full dose in seconds, there are also trans-effects on transcription and promoter activity and transcription synchronization, which are no longer present after 24 hours (Andrade-Lima et al., 2015;

PMID:25722371). Wildtype cells in culture supplied with the indicated levels of UV demonstrate complete transcription-coupled repair in a time course of 24 hours (see figure). The idea behind this analysis is not to demonstrate repair and recovery characteristics, which are already known for decades

in the DNA repair field, but to estimate the level of transcription-blocking DNA damage required to reduce transcription with 30%. For that aspect a wildtype cell line is not useful as repair has taken place.

Reviewer #2:

Remarks to the Author:

In the manuscript titled “Genome-wide transcription stalling by DNA damage shapes the transcriptome in aging”, the authors, Gyenis et al., report that age-associated transcriptional stress caused by DNA damage results in transcriptional stalling, altering the transcriptome in aged cells. They used novel sensitive nascent RNA sequencing to detect abnormalities in transcription in aged mouse liver and correlated them with RNA polymerase II stalling mapped by ChIP-seq. Finally, similar stalling and transcriptional defects were also detected in mouse models with defects in DNA repair, suggesting accumulated DNA damage during aging is likely the underlying cause.

Overall, this project is carefully designed and executed, most results are convincing and support the major conclusions. I find that a few more clarifications will make this manuscript acceptable for publication.

1. Consideration for EU labeling period and controls for optimization and labeling efficiency/equality between young and old animals.

We used a 5 hour labelling period, which was published by our colleagues to be able to label livers from premature aging mice for fluorescent imaging (Milanese et al., 2019; PMID: 31653834). This leads to a robust and consistent labeling of at least liver and kidney.

In addition, we added more proof about labeling efficiency / equality between young and old animals. We already observed that cDNA fragment length and length distribution (but not quantity) is almost identical in our protocol. Since reverse transcriptase cannot synthesize over an EU covalently bound by a biotin molecule, cDNA fragment size in itself already suggests similar EU incorporation frequencies (Extended data figure 2D). If there is a ~1.5-fold reduced availability of EU in the cellular rNTP pool, then there would be ~1.5-fold less incorporation of EU into nascent RNAs, i.e. the distance between incorporated EU is likely also 1.5-fold increased. As our EUSeq protocol isolates all nascent RNAs with at least one EU incorporated, a 1.5-fold reduction in EU incorporation should especially be visible in very small RNA species (<300 nucleotides) in which we would expect a decline in the percentage of reads mapping to those small RNA species. However, we measured a non-significant and modest increase in the percentage mapped reads in small RNA species in aging, indicating equal EU availability (Extended data figure 2). As a control, we also analyzed the percentage reads mapping to RNA species with a length of 1-3 kb or 2-4 kb. Also in these categories, we do not observe differences between young and old mice.

Next, we also calculated the probabilities that at least one EU is incorporated into a nascent RNA molecule for different EU incorporation rates and therefore available for sequencing. Note that EU-biotinylation is likely saturated as there is about >10x biotin available for every EU nucleotide in the reaction. The probability that 1.5-fold less available EU and concomitant less incorporation into small nascent RNA species results in similar percentages of mapped reads to those small RNA species is $p < 1 \cdot 10^{-16}$ (the lowest probability that R scripts can provide). As a control, we also calculated the probability that 1.5-fold less available EU in RNA species of 1-3 kb and 2-4 kb length, which was $p > 0.98$, indicating we would expect equal representation of these classes in the sequence data from young and old mice.

In conclusion, the scenario in which reduced EU availability leads to the observed results is highly unlikely.

2. Given the similarity in RNA polymerase II stalling between aged mice and mice with DNA repair defects (Xpg^{-/-} and Ercc1^{-/-}), it is reasonable to postulate that DNA damages are also causing RNA polymerase II stalling in aged mice. However, since DNA damages causing age-related RNAP II stalling is a key point in this manuscript, it should be further corroborated. For example, can DNA damage signals (like gamma-H2A.X) can be detected in RNAP II stalling sites in aged mice? Are DNA damage and RNAP II stalling

reduced in any of the longlived mouse models (either dwarf mouse models or known longevity interventions)?

This point raised is very understandable (and also raised by reviewer #1), but also technically much harder than generally thought. Unfortunately, outside of the DNA repair field there is a wide misunderstanding that accurately detecting physiological levels of endogenous DNA damage, that is highly heterogeneous in nature and each type of lesion present in minute quantities can be done. This is the reason we incorporated a separate paragraph in our recent Nature Review (Schumacher et al., 2021; PMID: 33911272) devoted to this topic which lists all the shortcomings with current technologies for quantitative detection of physiological DNA damage. While for few endogenous DNA lesions reliable assays have been developed, such as for double strand breaks by γ H2AX staining or for the oxidative damage 8-oxo-dG. While DSBs are present in very low quantities (<3 endogenous breaks / cell), which cannot explain the transcription stress phenotype, 8-oxo-dG lesions do not lead to transcription stalling. Hence, it is important to measure the correct type of damage. Moreover, in all likelihood there is not one type of DNA injury that is responsible for transcription stalling, but very probably a wide variety of very different types of lesions, including DNA-DNA and DNAprotein crosslinks and numerous types of distorting chemical modifications and adducts that interfere with proper base-pairing or physically block the polymerase. Finally, there is no method available that is able to determine the DNA lesion that actually blocks a RNA polymerase. The references cited in the manuscript quantify endogenous transcription-blocking lesions (e.g. cyclopurines, aldehyde damages, advanced glycation endproducts) that occur genome-wide in normal aging and demonstrate elevated levels in aging up to a point that they can explain our results.

One of the aims when we started the project leading to this manuscript was not only providing additional evidence for transcription-blocking lesions in normal aging, but also their functional consequences, i.e. that endogenous transcription-blocking DNA damage results in transcription stress. As DNA damage under normal physiological conditions is present in low quantities and heterogeneous in nature, there was a need to include an assay that detects all of them simultaneously, preferably in the gene body's template strand. Actually, this is why we are very happy with our assay, which measures the coding strand bias in aging, which is able to detect a wide variety of DNA lesions that interrupt polymerase activity and thus also transcription in the template strand of the transcribed compartment of the genome. The coding strand bias measurements are in itself already an assay detecting transcription-blocking DNA damage genome-wide.

In addition, we searched for DNA damage markers that would further provide evidence for transcription blocks by DNA damage and subsequent damage signaling. To our knowledge, the only known DNA damage signal that marks stalled transcription by DNA damage is non-canonical phosphorylation of DNA damage checkpoint kinase ATM, in which non-canonical refers to ATM activation in the absence of

double strand DNA breaks (Tresini et al., Nature 2015; PMID: 26106861). We tested non-canonical phosphorylation of ATM in young and old liver and found a significant increase in aging, which further substantiates our scenario of DNA damage-induced transcription stress in aging. We have added this data to Figure 4 and thank the referee for this comment.

We previously published that calorie restriction (CR), one of the most robust health- and lifespan-prolonging interventions indeed reduces DNA damage. CR enormously increases health- and lifespan of premature aging transcription-coupled DNA repair and global-genome nucleotide excision repair mutant mouse models, reduces markers of DNA damage such as γ H2AX and P53 staining, but importantly reduces the loss of long gene expression at the mRNA level (Vermeij et al., 2016; PMID: 27556946), indicating that CR protects against transcription stress as described in this manuscript. We have added this information to the discussion.

3. Figure 4h showed coding strand bias in aged mice? How do they compare to young mice?

We agree that figure 4h needed some additional clarification. The y-axis shows the fraction change over young mice. We have updated the text and legend for clarification.

4. In Figure 5a-b, the authors showed nascent RNA-seq profiles for Igf1 gene, making the point that transcriptional stress affects age-related pathways. It is also important to corroborate this point with RNAP II ChIP-seq data at Igf1 gene to directly show that RNAP II stalling is occurring in aged mice.

We have added RNAPII levels across the Igf1 gene body. As expected, RNAPII levels are elevated. In addition, we have also added the coding strand bias in aging effect. Interestingly, it correlates very well with the observed transcriptional decline as seen in EU-Seq and RNAPII level increase as seen in ChIP-Seq. The first part of the gene does not show a transcriptional decline or increase in RNAPII, but also no coding strand bias.

The last two thirds of the gene body exhibit transcriptional decline and increased RNAPII, correlating with a bias towards the coding strand, thereby further providing evidence that genomic perturbations, most likely DNA damage, is the cause of the observed transcription stress. This further strengthens the points made in the manuscript and we are grateful to the referee for this comment.

Reviewer #3:**Remarks to the Author:**

During the recent years, the research field on ageing has witnessed much and compelling evidence –mainly from the Hoeijmakers group- supporting the notion that persistent DNA damage drives aging. Further work revealed that the gradual accumulation of irreparable DNA lesions in the mammalian genome impinges selectively on pathways or processes that causally contribute to age-related diseases. However, the question remained how or why gene expression programs change when DNA damage accumulates over long periods of time, which genes or pathways are at greater risk or why certain types of cells or tissues are more vulnerable to persistent DNA lesions than others.

The present work is paradigm-shifting; it provides compelling, in vivo evidence that DNA damage interferes with the process of ongoing mRNA synthesis. The authors provide multiple evidence that transcription initiation remains unaltered with no difference on RNAPII recruitment on TSS or the transition of RNAPII from promoters to the main gene body. Strikingly, unlike the higher phosphorylated RNAPII levels and the functionally intact promoters, the authors find that nascent RNA synthesis gradually declines in highly transcribed genes with aging. The decline in gene transcription is dependent on primary gene length (and time). Using rapidly aging animals or mouse dermal fibroblasts with engineered defects in NER, the authors show that the severity of the DNA repair defect itself or the levels of oxidative or UV-induced DNA damage correlate well with the reduced transcription elongation. Importantly, the authors show that in old livers, there is coding strand bias, which also depends on gene length; that is, the DNA lesion in the template strand that stalls RNAPII also impairs DNA amplification of that strand. The authors then move on to provide evidence that their findings are universal and relevant to multiple species and organs with dramatically different developmental or functional requirements.

Overall, this compelling work should be published as is with no further experiments. The experiments are carefully designed and the conclusions are well supported from the primary observations. To further improve the content, the authors could reflect on whether exposure of mice to calorie restriction is expected to minimize the RNAP2 stalling frequencies in aged mouse livers.

We agree that it would be very informative to add a reflection about calorie restriction in the discussion. We initially omitted this due to word limitations. We previously showed that calorie restriction reduces DNA damage load and restores the loss of long gene expression at the mRNA level in premature aging mice with a transcription-coupled DNA repair defect (Vermeij et al., 2016; PMID: 27556946). Hence, a possible mechanism by which longevity might work is reducing DNA damage-induced transcription stress.

Decision Letter, first revision:

Our ref: NG-A59140R

27th Jul 2022

Dear Dr. Pothof,

Thank you for submitting your revised manuscript entitled "Genome-wide transcription stalling by DNA damage shapes the transcriptome in aging" (NG-A59140R). It has now been seen by the original referees and their comments are below. The reviewers find that the paper has improved in revision, and therefore we'll be happy in principle to publish it in Nature Genetics, pending revisions to satisfy reviewer #1's final requests and to comply with our editorial and formatting guidelines.

Since the current version of your manuscript is in a PDF format, please email us (natgen@us.nature.com) a copy of the file in an editable format (Microsoft Word)-- we can not proceed with PDFs at this stage.

We will then be performing performing detailed checks on your paper and will send you a checklist detailing our editorial and formatting requirements afterwards. Please do not upload the final materials and make any revisions until you receive this additional information from us.

Thank you again for your interest in Nature Genetics. Please do not hesitate to contact me if you have any questions.

Congratulations!

Sincerely,

Tiago

Tiago Faial, PhD
Senior Editor
Nature Genetics
<https://orcid.org/0000-0003-0864-1200>

Reviewer #1 (Remarks to the Author):

The authors have done a very nice job addressing most of the issues raised by us and other reviewers. I recommend the paper for publication at this point. However, I would like to make a few comments for the authors to consider. They should not be viewed as negative comments, but rather as suggestions for improving the narrative.

Starting from the abstract, the authors repeatedly stated that DNA damage is the cause of polII stalling toward the end of the genes and, hence, all the other phenotypes they observed. However, they did not directly show DNA damage (strand specific amplification is a good indication, but is still indirect and can have alternative explanations). The authors did not demonstrate polII stalling either (it can be slow moving polII). Furthermore, in the rebuttal letter they argue that the experiments to demonstrate DNA damage are too difficult to do. In my opinion, as the authors failed to explicitly demonstrate DNA damage, they should downplay it as a main reason for slowing down polII everywhere in the manuscript and particularly in the abstract. Unless the authors come up with a method to directly demonstrate DNA damage in vivo they should mention it as one of the possible reasons for the observed phenotypes.

Although the authors responded to most of my questions some of their answers were not satisfactory:

General issue 1. “However, low number of replicates for mRNA expression analysis experiments are quite commonly used, also in high impact journals.”

It is bad excuse to do a lower quality work if others did the same

Specific point 1. I understand it is difficult to “accurately detecting physiological levels of endogenous DNA damage”. Accordingly, I suggest to indicate that DNA damage may be one of the reasons for high polII occupancy toward the end of long genes.

Specific point 2. The authors’ response explains NTPs imbalance. But only the direct measurement of NTP concentration will provide a clear answer.

Specific point 3. “as figure 5a,b demonstrates is that sequence reads mapping to exons in the beginning of genes could also be increased, while a loss is observed in exons at the end of the gene body.” Same logic applies to introns as well. Therefore, this explanation is not satisfactory. The authors need to discuss this further in the text.

Reviewer #2 (Remarks to the Author):

The authors have addressed most of the concerns raised by reviewers. Overall, the manuscript has improved and provides new evidence for the roles of DNA damage in shaping the physiological outcome of aging. I recommend accepting this manuscript for publication.

Author Rebuttal, first revision:

Dear Editor,

We are pleased that you accepted our manuscript and that reviewer 2 has no additional comments. Please find our response to the reviewer 1.

Best regards,

Joris Pothof

General issue 1. “However, low number of replicates for mRNA expression analysis experiments are quite commonly used, also in high impact journals.”

It is bad excuse to do a lower quality work if others did the same

Response: increasing the number of replicates was not possible in a reasonable time frame and Nature Genetics was willing to overrule this point. To that end, it was recommended that we "responded to the concern about number of mice, potentially pointing to recent papers in the area to justify your experimental design", which we both did. Moreover, we added an additional mouse organ (kidney) to the manuscript with a similar transcription stress phenotype.

Specific point 1. I understand it is difficult to “accurately detecting physiological levels of endogenous DNA damage”. Accordingly, I suggest to indicate that DNA damage may be one of the reasons for high polIII occupancy toward the end of long genes.

Response: even though DNA damage is not directly demonstrated, it is -by far- the most logical interpretation.

Moreover, DNA polymerase apparently cannot amplify the purified template DNA strand, which implies that the DNA is chemically altered i.e. damaged, as normal DNA would have no problem to be amplified. In addition, this age-related coding strand bias increases towards the end of long genes, which we added in the revision. Furthermore, we ruled out many other reasons. We rephrased the discussion that we did not rule out all reasons that could explain the RNAPII occupancy toward the end of long genes, but think it is the most likely scenario.

Specific point 2. The authors' response explains NTPs imbalance. But only the direct measurement of NTP concentration will provide a clear answer.

Response: Although the reviewer is formally correct, NTP pool measurements require specialised HPLC procedures that we cannot carry out as we do not have the equipment nor the expertise. Moreover, we should then generate a new cohort of 2 year aged mice as this requires fresh liver tissues that need to be processed in a dedicated manner. While this would provide a definitive answer, a shortage of one NTP will only lead to transient slowing down transcription if that nucleotide is present in abundance compared to the other nucleotides. We ruled this out in our analyses. Moreover, an NTP imbalance cannot explain the observed age-related coding strand bias in RNAPII ChIP-Seq datasets, since this relies on in vitro amplification of native DNA in which all NTPs are abundantly present in equal amounts. The current text in our manuscript is worded in such a way that we do not exclude this possibility and does not require rephrasing.

Specific point 3. "as figure 5a,b demonstrates is that sequence reads mapping to exons in the beginning of genes could also be increased, while a loss is observed in exons at the end of the gene body." Same logic applies to introns as well. Therefore, this explanation is not satisfactory. The authors need to discuss this further in the text.

Response: The reviewer is correct that this is also observed in introns. However, the original specific point to which we responded concerned that also exons exhibit a gradual loss of transcription, which the reviewer missed. We sometimes see increased expression in the beginning of genes, but could not detect a robust pattern across genes and is not always present in genes with transcription stress (see e.g. the Ghr gene in Figure 4j). Since there could be several explanations and we currently do not know the underlying biological cause, we would like to refrain from describing this phenomenon in this manuscript as we think it will be distracting and word count limits may not allow it.

Final Decision Letter:

7th Dec 2022

Dear Joris,

I am delighted to say that your manuscript entitled "Genome-wide RNA polymerase stalling shapes the transcriptome during aging" has been accepted for publication in an upcoming issue of Nature Genetics.

Your paper will be published online after we receive your corrections and will appear in print in the next available issue. You can find out your date of online publication by contacting the Nature Press Office (press@nature.com) after sending your e-proof corrections. Now is the time to inform your Public Relations or Press Office about your paper, as they might be interested in promoting its publication. This will allow them time to prepare an accurate and satisfactory press release. Include your manuscript tracking number (NG-A59140R1) and the name of the journal, which they will need when they contact our Press Office.

Please note that *Nature Genetics* is a Transformative Journal (TJ). Authors may publish their research with us through the traditional subscription access route or make their paper immediately open access through payment of an article-processing charge (APC). Authors will not be required to make a final decision about access to their article until it has been accepted. [Find out more about Transformative Journals](https://www.springernature.com/gp/open-research/transformative-journals)

Authors may need to take specific actions to achieve > **compliance with funder and institutional open access mandates**. If your research is supported by a funder that requires immediate open access (e.g. according to [Plan S principles](https://www.springernature.com/gp/open-research/plan-s-compliance)) then you should select the gold OA route, and we will direct you to the compliant route where possible. For authors selecting the subscription publication route, the journal's standard licensing terms will need to be accepted, including <https://www.nature.com/nature-portfolio/editorial-policies/self-archiving-and-license-to-publish>. Those licensing terms will supersede any other terms that the author or any third party may assert apply to any version of the manuscript.

Please note that Nature Portfolio offers an immediate open access option only for papers that were first submitted after 1 January, 2021.

Sincerely,

Tiago

Tiago Faial, PhD
Chief Editor
Nature Genetics
<https://orcid.org/0000-0003-0864-1200>

Click here if you would like to recommend Nature Genetics to your librarian
<http://www.nature.com/subscriptions/recommend.html#forms>